



# Dheed: an ERA5 based global database of dry and hot extreme events from 1950 to 2022

Mélanie Weynants[1], Chaonan Ji[2], Nora Linscheid[1], Ulrich Weber[1], Miguel D. Mahecha[2], and Fabian Gans[1]

[1]Max Planck Institute for Biogeochemistry, Jena, Germany
[2]Remote Sensing Center for Earth System Research, Leipzig University, Germany

**Correspondence:** Mélanie Weynants (mweynants@bgc-jena.mpg.de); Fabiam Gans (fgans@bgc-jena.mpg.de)

**Abstract.** The intensification of climate extremes is one of the most immediate effects of global climate change. Heatwaves and droughts have uneven impacts on ecosystems that can be exacerbated in case of compound events. To comprehensively study these events, e.g. with local high-resolution remote sensing or in-situ data, a global catalogue of such events is essential. Here, we propose a workflow to build a database of large-scale dry and hot extreme events based on data from ERA5 reanalysis. Drought indicators are constructed based on evapotranspiration and precipitation data averaged over 30, 90 and 180 days. Extreme events are detected with the peak-over-threshold approach for the 1950–2022 period. Extremes and non-extremes are defined for daily maximum temperature at 2 m in combination with three drought indicators. In the last step, the spatiotemporal extent of the events is computed by a connected component analysis. The identified events are validated against extreme events reported in the literature.

## 1 Introduction

Extreme weather and climate events can induce stress on ecosystems and thereby have negative impacts on society, e.g. via yield losses with unclear implications (Frank et al., 2015; Sippel et al., 2018; Mahecha et al., 2024). With Earth climate currently changing, the intensity and frequency of heat and hydroclimatic extremes are increasing (Seneviratne et al., 2023; Rodell and Li, 2023). Specifically, concerns about compound extreme weather and climate events – when multiple types of climate extremes occur simultaneously – have been raised for over a decade (IPCC, 2012). A typology to guide studies on those types of occurrences has recently been proposed (Zscheischler et al., 2020). Compound climate extremes often have more detrimental effects on vegetation growth than univariate extremes (Yang et al., 2023; Bastos et al., 2023). For instance, global increased drought and heat (DH) induced tree mortality has been highlighted in 2010 (Allen et al., 2010) and investigated ever since. Vegetation is indeed more susceptible to damage during heat extremes after exposure to drought stress, as less water is available to buffer the physiological consequences of the heat extreme (Marchin et al., 2022). The complex physiological mechanisms of increased tree mortality under a warming and drying atmosphere richer in $CO_2$ are, however, still debated (McDowell et al., 2022). The cascading processes triggered by concurrent DH extremes also impact society as a whole (Niggli et al., 2022), and require particular focus given the expected increasing burden on society by DH in many parts of the world.



To study the impacts of dry and hot extreme events (DHEE) globally, a unified database of such events is needed. Yet, the
definition of heatwaves and droughts is not standardized in the literature. The World Meteorological Organisation describes
heatwaves as "periods where local excess heat accumulates over a sequence of unusually hot days and nights" (https://wmo.
int/topics/heatwave), but it defines no universal indicator. The scientific literature abounds with heatwave indicators, often
sector oriented (Perkins and Alexander, 2013). Many define a heatwave as a period of at least three consecutive days with
maximum temperature exceeding a certain threshold (e.g., Perkins and Alexander, 2013; Russo et al., 2015; Lavaysse et al.,
2018; Russo and Domeisen, 2023), either absolute or percentile based. These probabilistic thresholds can be regional or local
and relative to reference periods ranging from calendar day to season or year, over spans of ten to thirty years. Droughts
are prolonged dry periods that can last from weeks to years. Depending on their duration and intensity, the hydrological
compartment affected differs, and so does the impact on ecosystems. One generally distinguishes between meteorological,
hydrological, agricultural and socio-economic droughts (Mishra and Singh, 2010). Various indicators have been developed
to characterize drought conditions. The commonly used Standard Precipitation Evaporation Index (SPEI) is a "multi-scalar
drought index used to determine the onset, duration and magnitude of drought conditions" (Vicente-Serrano et al., 2010).
It is generally calculated from monthly climate data, but some authors have used it with daily data to characterize drought
dynamics at a finer temporal resolution (Wang et al., 2021). Indeed, Li et al. (2021) highlight the need for sub-monthly scale
indices to monitor short term compound dry and hot conditions. The rationale is that, otherwise, e.g. a four week drought
happening across two months might remain undetected in monthly data. A recent study proposes to calculate the daily SPEI
using nonparametric Kernel Density Estimation (KDE) and then transform the KDE based quantiles into standardized normal
scores, thereby avoiding fitting a parametric distribution to the data (Pohl et al., 2023). As sub-monthly dry and hot conditions
– or even a few hot days – can propagate into impacts, and heatwaves and droughts evolve on different time scales, we find
it advantageous to work on data with daily resolution, and with multi-scalar drought indicators representing water budgets for
different temporal windows.

Studies have often focused on single compound events (Flach et al., 2018; Ciais et al., 2005; Bastos et al., 2020, e.g.,), on
specific regions or measurement stations (Li et al., 2021; Pohl et al., 2023, e.g.,), but to the best of our knowledge no global
gridded analysis of DHEE at daily scale has been published so far. Therefore, here we propose Dheed, a database of large-scale
dry and hot extreme events derived from the analysis of long time series of the ERA5 global climate reanalysis provided by the
European Centre for Medium Range Weather Forecasts (ECMWF) (Hersbach et al., 2020, 2023). Dheed provides the potential
for analyzing historical patterns and trends in heatwaves and droughts over time and facilitates detailed studies on the impact
of dry and hot climate extremes on ecosystems, species, and regions. For example, it can serve for driving the sampling of
minicubes of high-resolution satellite imagery – e.g., Copernicus Sentinel-2 data (Ji et al., 2024) – to train models predicting
ecosystem states (Requena-Mesa et al., 2021; Benson et al., 2023) under extreme climate conditions. Dheed can also be used to
assess the capacity of ecological monitoring networks to detect impacts of dry and hot extreme events (Mahecha et al., 2017).
Further potential applications encompass site selection for studying the effects of extreme dry and hot conditions on specific
species or targeted sampling of high-resolution Earth Observation data for impact research, e.g. assess and forecast carbon
sequestration loss during extremes, or cropland productivity loss. Hereafter, we describe the data and methods employed to



build Dheed, we present a brief global and continental analysis of trends in DHEE and we benchmark detected DHEE against
events reported in the literature.

## 2    Data and methods

Our approach draws on the concept of analysis-ready data cube, particularly useful in Earth system science to access and
analyse multiple data dimensions, such as variable, spatial and temporal (Mahecha et al., 2020). The first step in building
Dheed consists of a temporal analysis of climate reanalysis data to detect extreme values in time series of temperature and
drought indices, which we further refer to as Discrete Extreme Occurrences (DEOs). It is followed by a spatio-temporal
connected component analysis to group DEOs connected in space and time into uniquely labelled Dry and Hot Extreme Events
(DHEEs) (Zscheischler et al., 2013; Lloyd-Hughes, 2012) that can then be compared to events reported in the literature. The
workflow, detailed below and illustrated in Figure 1, runs entirely in Julia, relying largely on the YAXArrays.jl package (Gans
et al., 2023). All figures are created with Makie.jl (Danisch and Krumbiegel, 2021).

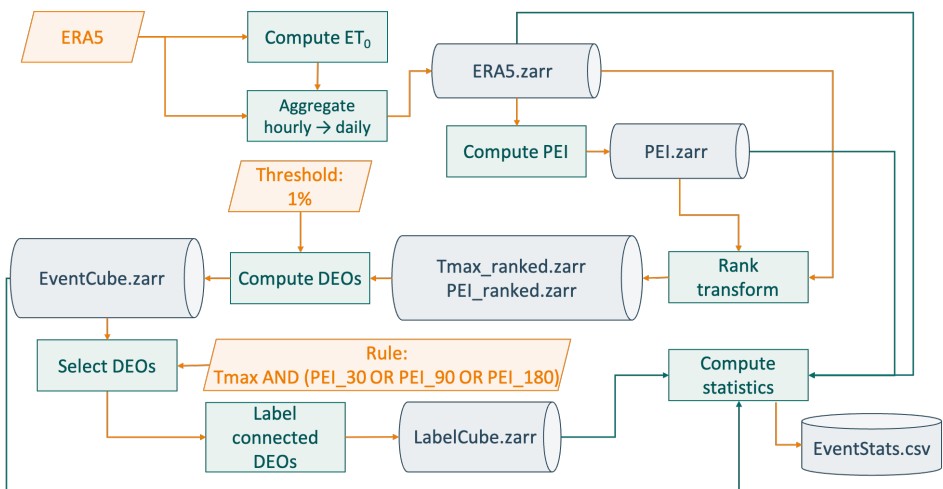

**Figure 1.** Workflow for the detection of dry and hot extreme events. $ET_{ref}$ is the reference evapotranspiration, PEI is a Precipitation–
Evapotranspiration based indicator, Tmax is the daily maximum temperature at 2 m. DEOs are Discrete Extreme Occurrences, i.e. extreme
values in the time series of temperature and drought indices. Dheed, the resulting dry and hot extreme events database consists of the
EventCube, the LabelCube and the EventStats table.

## 70    2.1    Climate data pre-processing

The workflow exploits the hourly gridded ERA5 data, from 1950 to 2022 (Hersbach et al., 2023). Specifically, the following
variables are used: temperature at 2 meters ($T_{2m}$) [K], 10 meter wind speed: zonal ($u10$) and meridional ($v10$) components
[ms$^{-1}$], atmospheric surface pressure ($sp$) [Pa], surface net solar and thermal radiation ($ssr$ and $str$) [Jm$^{-2}$], saturation



water vapour pressure ($swvp$) [hPa], vapour pressure ($vp$) [hPa] and total precipitation ($tp$) [m]. Grid cells from the ERA5
land mask with a value greater than 0.5 are considered land. Data are aggregated over time from hourly to daily time steps, by
calculating the daily mean, minimum, and maximum for $T_{2m}$, and the cumulative values for $tp$ and $ET_{ref}$ (see hereafter). When
aggregating to daily time steps, a day includes all time steps from 0:00 to 23:00 UTC for all grid cells. Hence, aggregation
windows do not correspond to local calendar days. The resulting data are stored in a data cube in Zarr format with the following
chunk sizes: longitude = 60, latitude = 60, time = 5,844, suited for time series analysis. As in the original ERA5 data, the
80 longitude axis ranges from 0 to 360 degrees. The spatial resolution is 0.25 degree in both latitude and longitude, i.e., the
longitude and latitude dimensions are 1,440 and 721 respectively. After aggregation of the hourly data to daily temporal
resolution, each time series has 26,663 data points over the period going from 1 January 1950 to 31 December 2022.

The hourly reference evapotranspiration for a well watered grass cover ($ET_{ref}$) [mm hr$^{-1}$] is calculated with the FAO's
Penman-Monteith equation (Allen et al., 1998) from the above mentioned ERA5 variables, following appropriate units adjust-
85 ments and assumptions (Singer et al., 2021):

$$ET_{ref} = \frac{0.408\Delta(R_n - G) + \gamma\frac{37}{\theta_{2m}+273}u_2(swvp - vp) \times 10^{-1}}{\Delta + \gamma(1 + C_d u_2)} \tag{1}$$

where $R_n$ is the surface net radiation [MJm$^{-2}$hr$^{-1}$]), calculated as $(ssr + str) \times 10^{-6}$, $G$ is the soil heat flux density at the
soil surface [MJm$^{-2}$ hr$^{-1}$] conditioned on the time step, with values differing between daytime and nighttime ($G = R_n \times 0.1$ if
$R_n < 0.0$, $G = R_n \times 0.5$ otherwise) and set to 0 where water covers more than 50% of the spatial grid cell, $\theta = T_{2m} - 273.15$
is the mean daily air temperature at 2 m height [°C], $u_2$ is the wind speed at 2 m height [m s$^{-1}$], calculated from $u_{10}$ and
$v_{10}$ using the log wind profile (Equation 2), $\Delta$ is the slope of the vapour pressure curve [kPa°C$^{-1}$], $\gamma$ is the psychrometric
constant [kPa°C$^{-1}$] and $C_d$ is a time step dependent coefficient. $\Delta$ and $\gamma$ are calculated from $sp$ and $\theta_{2m}$ according to FAO
recommendations (Allen et al., 1998) with equations 3 and 4.

$$u_2 = \sqrt{u_{10}^2 + v_{10}^2}\frac{4.87}{\log(67.8 \times 10 - 5.42)} \tag{2}$$

$$\Delta = 4098.0\frac{0.6108\exp\frac{17.27\theta_{2m}}{\theta_{2m}+237.3}}{(\theta_{2m}+237.3)^2} \tag{3}$$

$$\gamma = cp\frac{sp}{\epsilon\lambda} \tag{4}$$

where $\lambda = 2.45$ is the latent heat of vaporization [MJ kg -1] (simplification in the FAO PenMon (latent heat of about 20°C),
$cp = 1.013 \times 10^{-3}$ is the specific heat at constant pressure [MJ kg-1 °C-1] and $\epsilon = 0.622$ is the ratio between molecular weight
of water vapour and dry air. According to Walter et al. (2001), $C_d$ should vary between daytime (0.24) and nighttime (0.96),
but adopting the constant value for daily calculation (0.34) has a negligible effect when values are aggregated by day ($< 10^{-6}$).



For the purposes of this dataset, daily drought conditions need to be assessed to be later analyzed together with a daily
heatwave indicator to detect extreme events at daily resolution (Li et al., 2021). Therefore, the daily average water balance
$\text{PEI}_{N,i}$ for day $i$ in the time series over the $N$ antecedent days is calculated as an indicator of drought, accounting for different
hydrological compartments of ecosystems:

$$\text{PEI}_{N,i} = \frac{1}{N} \sum_{j=i-N-1}^{i} (tp_j \times 10^{-3} - \text{ET}_{ref,j}) \tag{5}$$

with $N \in (30, 90, 180)$ to obtain $\text{PEI}_{30}$, $\text{PEI}_{90}$, and $\text{PEI}_{180}$. Following the convention used in ERA5, downward fluxes have
positive values. Extreme dry values are hence those for which $\text{PEI}_N$ is small.

## 2.2 Event detection

The detection of DEOs is based on a purely probabilistic threshold applied to the empirical distribution of the indicators. In
each spatial grid cell, we examine the temporal distribution of each of the four indicators independently ($T_{max}$, $\text{PEI}_{30}$, $\text{PEI}_{90}$,
and $\text{PEI}_{180}$). Fitting a parametric distribution to the time series of the drought indicators to obtain a standarized index (SPEI)
proved difficult for many grid cells. Instead, the values were rank-transformed to obtain their empirical distribution function,
as an estimate of the cumulative distribution function of each spatial grid cell. We applied the same rank-transformation to
$T_{max}$ values after reversing them so that the extremes of interest are consistently the smallest values for all four indicators.
We choose an absolute threshold specific to each grid cell to focus on extreme hot conditions, and do not consider here winter
heatwaves. We set the lowest 1% of the empirical cumulative distribution as extreme. We synthesize the DEOs of the four
indicators in a single variable encoded as an 8-bit integer by assigning a specific bit to each indicator. DEOs of $T_{max}$ activate
the first (smallest) bit, $PEI_{30}$ the second, $PEI_{90}$ the third, and $PEI_{180}$ the fourth. The fifth bit encodes for all values that
lie outside the tails of all four distributions, i.e., that have values between 0.1 and 0.9. These values are stored in a data cube
named `EventCube` (Fig. 1).

From the `EventCube`, we extract DHEEs as labelled groups of connected dry and hot DEOs (Zscheischler et al., 2013;
Lloyd-Hughes, 2012). We restrict the analysis to spatio-temporal grid cells of the `EventCube` that have uneven values greater
than 1, i.e. DEOs that are flagged as both hot and dry extremes. Moreover, using *ImageFiltering.jl* (Ima, 2023a), we filter
for events that last at least three consecutive days. The connected component labelling algorithm assigns a unique label to
each group of connected DEOs, looking for six way connections. Each grid cell with coordinates $(x \pm 1, y, z)$, $(x, y \pm 1, z)$ or
$(x, y, z \pm 1)$ is connected to the grid cell at $(x, y, z)$, with $x$, $y$ and $z$ the longitude, latitude and time, respectively. We modify the
*ImageMorphology.label_components* function (Ima, 2023b) to group DEOs connecting across the globe along the longitude
dimension. We also introduce the possibility to merge labels from contiguous data cubes. We store the resulting labelled dry
and hot extreme events in a data cube named `labelCube`. Figure 2 illustrates the entire workflow with the example of the 2003
summer heatwave in Europe (Event 33 in Table A1.

For each labelled DHEE, we compute the following properties:





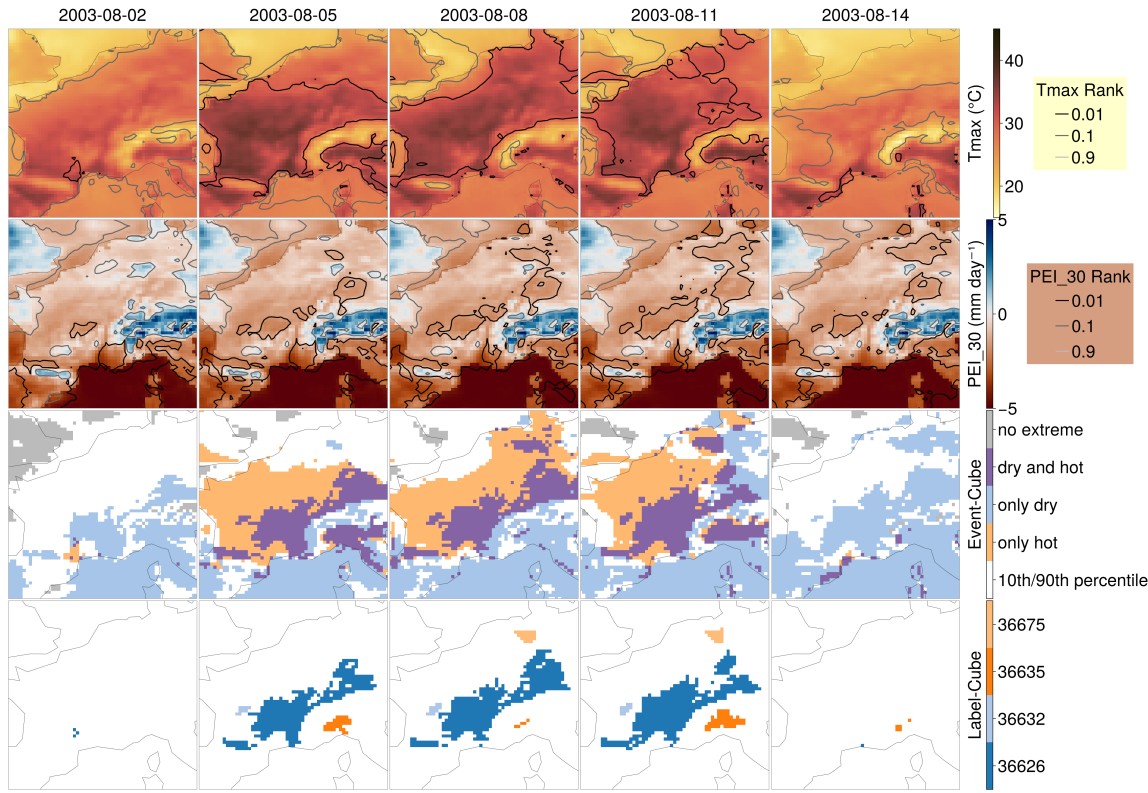

**Figure 2.** Example of dry and hot extreme event detection workflow over the 2003 summer heatwave in Europe. Columns show the time evolution of the data sampled at every 4th time step from Aug 1 to Aug 13 2003. Row 1 and 2 show the raw daily maximum 2m air temperature with isolines for the grid cell based 1%, 10% and 90% threshold. The third row shows the encoding into the EventCube where single voxels can be marked as only extremely dry, only extremely hot, a combination of both or none of them, while voxels plotted as white are in a regime of normal conditions between 0.1 and 0.9 percentile. Through a connected component analysis in space and time these voxels are connected into four connected components that are registered as separate labelled events in the Dheed dry and hot extreme events database.

- spatio-temporal bounding box (`start_time`, `end_time`, `longitude_min`, `longitude_max`, `latitude_min`, `latitude_max`),

- statistics on the the indicators (`t2mmax_mean`, `t2mmax_min`, `t2mmax_max`, `pei_30_mean`, `pei_30_min`, `pei_30_max`, `pei_90_mean`, `pei_90_min`, `pei_90_max`, `pei_180_mean`, `pei_180_min`, `pei_180_max`),

- percentage of the event affected by a single indicator (`heat`, `drought30`, `drought90`, `drought180`, `compound`),

- percentage of the event that occurred over land (`land_share`),

- a proxy of the total volume of the event as the number of voxels weighted by cos(latitude) (`volume`),





  – the event duration as $\texttt{end\_time} - \texttt{start\_time} + 1\,\text{day}$ (`duration`),

  – a proxy of the event total affected area as the ratio between `volume` and `duration` (`area`).

These statistics are stored in a csv table named EventStats (Fig. 1) and constitute the core of Dheed, along with EventCube
and labelCube. In the next section, we present a brief analysis of these labelled DHEEs and track the ten largest in `volume`
and the ten longest in `duration` in the scientific literature. To assess the reliability of the event detection method, we also
compare a set of historical events reported in the scientific literature or the media with the Dheed. All labelled events that
intersect with the spatio-temporal window reported in Table A1 are selected from the labelCube. Their statistics are extracted
from the EventStats and evaluated.

## 3 Results

### 3.1 Indicators of dry and hot conditions

All detected daily time points of extreme heat or drought (DEOs) from 1950-2022 are recorded in the EventCube. This data
cube can be used to analyse time series of DEOs at specific locations. For example, Figure 10 shows the event type along with
$T_{2,max}$ from the ERA5 daily data cube and the PEIs for a few days in the summer of 2021 at Lytton, British Columbia, Canada.
Longer time series can also be easily analysed (Figures A1 and A2). Beyond analysing single locations, the dataset allows to
draw a general overview of the regional or global trends in dry and hot extremes. Figure 3 shows DEOs globally aggregated
by year and by type of extreme, over land only, from 1970 to 2022. The percentage is the sum of land grid cells weighted
by the cosine of their latitude, representing the annual number of days × area, reported on the total number of land grid cells
multiplied by the cosine of their latitude in that year, representing the total land area multiplied by the number of days in the
year. The further back in time, the larger the uncertainties in the reanalysis data, due to a lack of observations to be assimilated
with the numerical model results, especially in the southern hemisphere. No satellite data were used in ERA5 before 1970
(Hersbach, 2023), leading to yet larger uncertainties in the southern hemisphere. Therefore, we do not include the years 1950–
1969 in the analysis. Figure 3 shows DEOs by event type aggregated globally over land by year. The compound dry and hot
DEOs shown in purple represent only a small fraction of the extreme dry or hot conditions. The inter-annual variability is large,
but there seems to be a positive trend. Figure 4 shows the DEOs aggregated by macro type of extreme, which means the bars
may not be cumulated. By our definition of the extremes, since we applied a 1% threshold on the time series, the relative annual
number of days and area affected by these extremes expressed in percentage for each macro type is 1% on average over the
complete time series (1950–2022). Values vary however from year to year. A Theil-Sen approximation of the trend over time
shows that all three drought indicators have a positive trend. Surprisingly the trend for maximum temperature extremes is not
significant (p–value of Mann–Kendall test greater than 0.05). This suggests that the extents of short term (d30), midterm (d90)
and longer term (d180) droughts are increasing at a similar pace. Figure 5 shows only grid cells affected by compound dry and
hot extremes aggregated globally by year. The values are an order of magnitude smaller than with the univariate extremes, but
the trend is such that there was a tripling of the annual affected area over the 53 years investigated. Figure 6 and Table 1 show

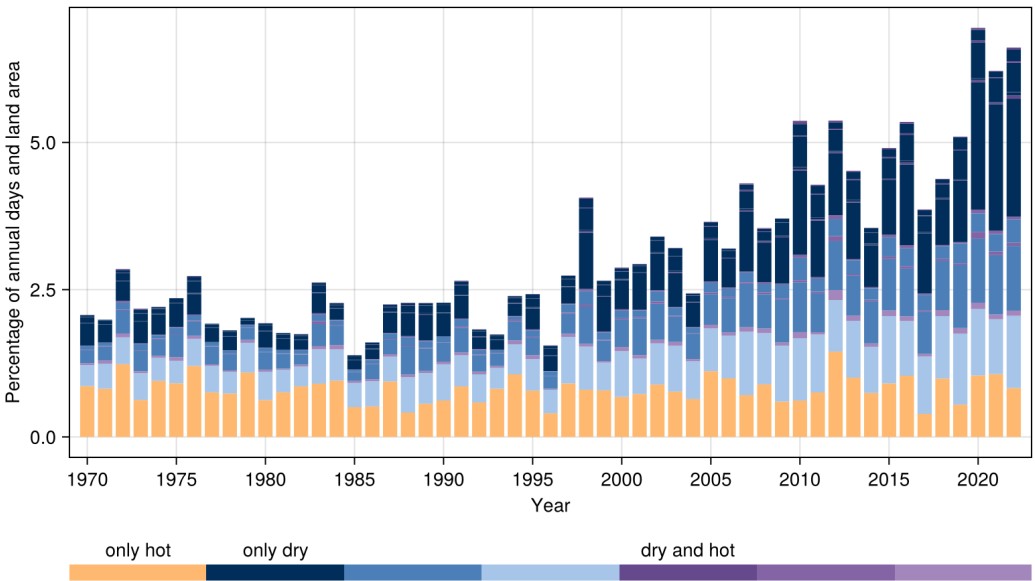

**Figure 3.** Annual spatiotemporal extent of extreme dry and hot days, by the value of data in EventCube. The sum of Discrete Extreme Occurrences (DEO) of a given value weighted by the cosine of the grid cell latitude is divided by the sum of all land voxels in a given year, expressed as percentage.

a continental aggregation of the annual `volume` affected by compound dry and hot DEOs. The trends and average values are not uniform across continents. With 0.03 %, Antarctica is the least affected continent, well below the global average of 0.13 %. Australia and South America are below average with 0.10 and 0.11 %. However, they saw a steep increase of extreme dry and hot days in recent years, reaching averages close to or above the global average for the period 2000-2022. Europe is the continent most affected by compound dry and hot events, followed by Africa. Africa is the continent that has experienced the steepest increase in annual cumulative area subject to compound extreme dry and hot days.

## 3.2 Database of dry and hot extreme events

Extreme events in which heat and drought coincided were labelled and characterized further. The labelled extreme events include only DEOs where dry and hot extreme conditions were observed for at least three consecutive days. Although the connected components algorithm was run over all grid cells, the statistics for the labelled events were computed over land only. In total, the database contains 26,351 unique labelled events for the years 1970 to 2022. Most events have a `duration` of four days and a spatial extent smaller than a grid cell at the equator (`area`) (Fig. 7, left). In recent years, there were more dry and hot extreme events (Fig. 7, right). The distribution of their `volume` remains however stable over time. Nevertheless, seven of the ten largest events occurred after the year 2000. They are listed in Table 2 (top) along with the ten longest events (bottom).

**Table 1.** Average percentage of annual extremely dry and hot days and area by continent and globally, over the total analysis period (1970-2022), over the first 30 years (1970-1999) and over the recent years (2000-2022).

| Continent | Years_1970_2022 | Years_1970_1999 | Years_2000_2022 |
|---|---|---|---|
| Africa | 0.15 | 0.06 | 0.26 |
| Asia | 0.14 | 0.11 | 0.17 |
| Australia | 0.10 | 0.04 | 0.17 |
| North America | 0.14 | 0.11 | 0.19 |
| Oceania | 0.13 | 0.10 | 0.18 |
| South America | 0.11 | 0.05 | 0.20 |
| Antarctica | 0.03 | 0.02 | 0.04 |
| Europe | 0.21 | 0.13 | 0.31 |
| **Global** | 0.13 | 0.08 | 0.19 |

**Table 2.** Biggest labelled dry and hot extreme events in the period 1970–2022: Ten largest in `volume` and ten longest in `duration`.

| | | Date | | Longitude | | Latitude | | | | |
|---|---|---|---|---|---|---|---|---|---|---|
| rank | label | start | end | min | max | min | max | duration | area | volume |
| 1 | 42561 | 2010-07-12 | 2010-08-21 | 28.00 | 63.25 | 36.25 | 64.25 | 41 days | 1640.94 | 67278.7 |
| 2 | 51252 | 2016-10-07 | 2016-11-04 | 13.50 | 34.50 | -24.00 | -5.75 | 29 days | 1266.11 | 36717.1 |
| 3 | 24092 | 1983-03-01 | 1983-03-30 | 0.00 | 359.75 | 1.75 | 10.25 | 30 days | 685.372 | 20561.2 |
| 4 | 55983 | 2021-02-20 | 2021-03-16 | 16.50 | 31.25 | -2.50 | 10.00 | 25 days | 698.638 | 17465.9 |
| 5 | 55755 | 2020-09-30 | 2020-10-14 | 288.25 | 316.75 | -23.25 | -10.75 | 15 days | 980.048 | 14700.7 |
| 6 | 25632 | 1986-07-02 | 1986-07-22 | 102.00 | 135.25 | 54.75 | 65.25 | 21 days | 609.035 | 12789.7 |
| 7 | 18958 | 1972-08-18 | 1972-08-30 | 31.00 | 51.50 | 47.75 | 60.75 | 13 days | 763.996 | 9931.95 |
| 8 | 44770 | 2012-06-17 | 2012-07-04 | 247.75 | 259.50 | 26.75 | 41.75 | 18 days | 485.379 | 8736.83 |
| 9 | 36790 | 2003-10-03 | 2003-10-19 | 18.50 | 31.25 | -21.50 | -6.25 | 17 days | 486.031 | 8262.52 |
| 10 | 53015 | 2019-01-11 | 2019-01-26 | 123.25 | 149.00 | -36.0 | -23.75 | 16 days | 510.833 | 8173.33 |
| 1 | 31767 | 1998-03-01 | 1998-04-27 | 115.00 | 118.50 | -1.25 | 3.25 | 58 days | 67.1294 | 3893.51 |
| 2 | 31866 | 1998-03-07 | 1998-04-30 | 115.00 | 118.50 | 4.00 | 6.50 | 55 days | 20.7583 | 1141.71 |
| 3 | 49981 | 2015-09-13 | 2015-10-26 | 123.00 | 124.75 | 0.50 | 1.25 | 44 days | 7.43092 | 326.96 |
| 4 | 44843 | 2012-06-23 | 2012-08-04 | 229.25 | 272.25 | 69.75 | 78.50 | 43 days | 47.4123 | 2038.73 |
| 5 | 18998 | 1972-09-10 | 1972-10-21 | 109.25 | 116.25 | -3.75 | -0.25 | 42 days | 85.2228 | 3579.36 |
| 6 | 24109 | 1983-03-01 | 1983-04-11 | 114.50 | 118.50 | -2.00 | 4.25 | 42 days | 58.1754 | 2443.37 |
| 7 | 42561 | 2010-07-12 | 2010-08-21 | 28.00 | 63.25 | 36.25 | 64.25 | 41 days | 1640.94 | 67278.7 |
| 8 | 28223 | 1992-03-25 | 1992-04-30 | 98.50 | 105.00 | 12.75 | 20.25 | 37 days | 135.121 | 4999.47 |
| 9 | 50340 | 2016-02-27 | 2016-04-02 | 283.50 | 288.00 | 6.75 | 11.00 | 36 days | 57.2994 | 2062.78 |
| 10 | 54071 | 2020-03-06 | 2020-04-09 | 80.00 | 80.75 | 6.00 | 7.50 | 35 days | 8.14539 | 285.089 |

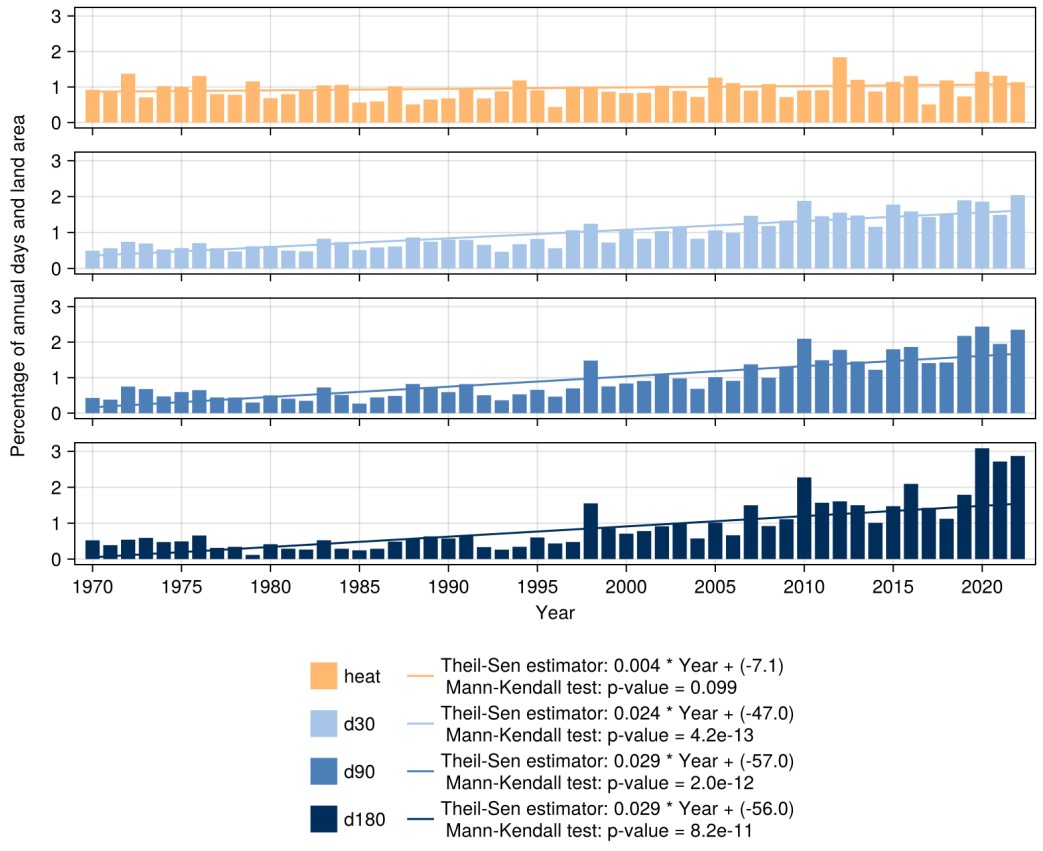

**Figure 4.** Annual spatiotemporal extent of extreme dry and hot days, by type of extreme. The sum of Discrete Extreme Occurrences (DEO) combined by type of event and weighted by the cosine of the grid cell latitude is divided by the sum of all land voxels in a given year, expressed as percentage.

Figure 8 shows the ten largest labelled events that occurred in the years 1970 to 2022. The largest event overall – labelled 42561 – relates to the Russian heatwave of 2010 (e.g. Flach et al., 2018). It also appears as the seventh longest dry and hot event globally over the same period (Table 2, bottom). Events labelled 51252 (second largest) and 36790 (ninth largest) can be linked to the El Niño induced droughts in southern Africa in 2002–2003 and 2015–2016 Rouault et al. (2024). Event labelled 24092 in Western and central Africa in 1972 can be traced back to the extreme drought of 1972–1973 (Masih et al., 2014). The serious drought of 2021 in Central Africa, which compounded the humanitarian situation caused by COVID-19, internal conflicts, and plague locusts in the region (Hassan et al., 2023), is captured in Event 55983. A severe drought-complex hit over the Pantanal and other regions in South America in October 2020, increasing fires and impacts on natural and human systems (Marengo et al., 2022), to which Event 55755 can be associated. In the spring and summer of 2012, the United States of America suffered through their hottest year on record, which complicated and exacerbated the ongoing drought situation

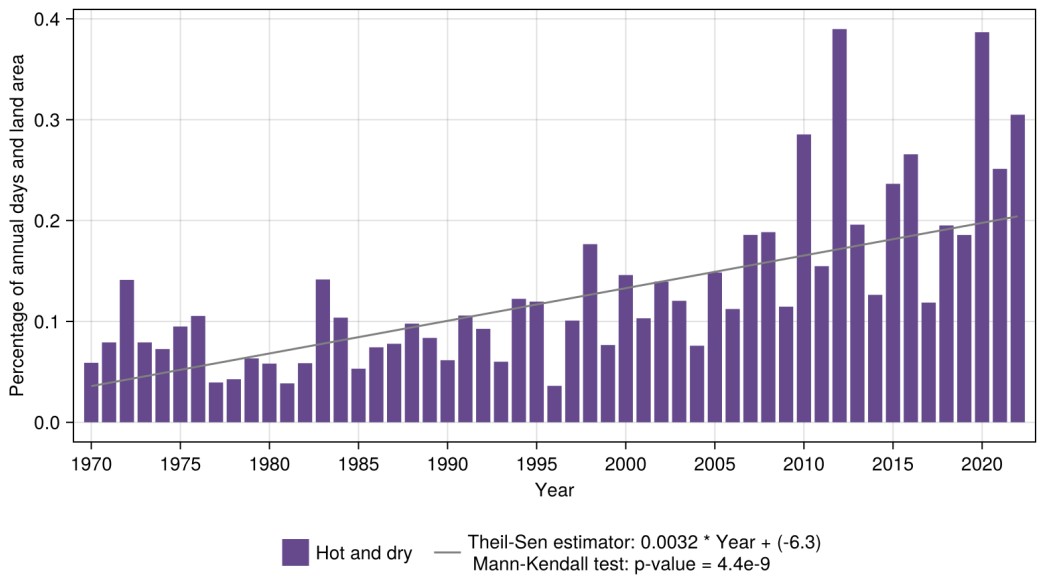

**Figure 5.** Annual spatiotemporal extent of extreme compound dry and hot days. The sum of Discrete Extreme Occurrences (DEO) that are both dry and hot weighted by the cosine of the grid cell latitude is divided by the sum of all land voxels in a given year, expressed as percentage.

(Rippey, 2015). Event 53015 occurred in January 2003, which was Australia's hottest during the country's hottest summer on record (http://www.bom.gov.au/climate/current/month/aus/archive/201901.summary.shtml, Accessed 12 August 2024). No references could be found to Events 25632 and 18958, which occurred in Siberia in July 1986 and Russia in 1972, respectively.

El Niño–induced droughts in Malaysia/Indonesia (Borneo-Kalimantan Island) in 1972-1973, 1982-1983, 1997-1998, 2014-2016 are captured in the longest events 18998, 24109, 31767, 31866, 49981 (Table 2, bottom), which contributed to triggering increased forest mortality (Allen et al., 2010). The long event 44843 in the Canadian-Arctic Archipelago in the summer of 2012 can be linked to the summer heatwave that affected North America. A mega-drought was reported in Colombia in 2015-2016, which was accompanied by a strong warming in the Amazon forest (Weng et al., 2020) and can be linked to the long event 50340 that occurred in February-April 2016. No specific references were found to the long events 28223 in Mainland Southeast Asia (Indochina Peninsula) in 1992 and 54071 in Sri Lanka in March 2020.

### 3.3 Validation

Next to the largest and longest extreme dry and hot events discussed in the previous section, the database was tested against a list of extreme events gathered independently and a priori (Table A1). The intersection of the reported approximate footprint and time range of those events with the database proposed here is summarized in Figure 9. Reported events are generally associated with a few large labelled events and with many small labelled events. This is consistent with the distribution of the



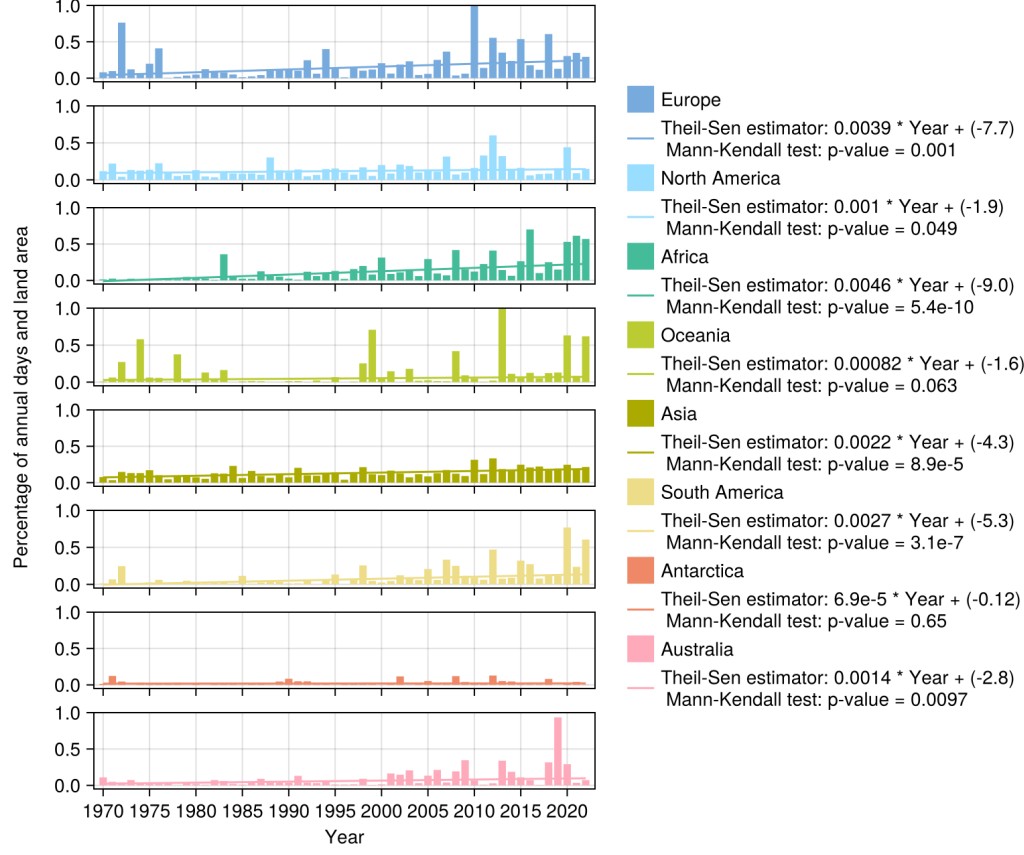

**Figure 6.** Annual spatiotemporal extent of extreme dry and hot days, by continent. The sum of both dry and hot Discrete Extreme Occurrences (DEO) weighted by the cosine of the grid cell latitude is divided by the sum of all land voxels in a given continent and year, expressed as percentage. The y axis is limited to 1.0 %, but the bar for the year 2010 in Europe extents to 1.8 % and that of the year 2013 in Oceania to 1.1 %.

size of the labelled events (Figure 7). However, some reported events (8, 25, 28, 31, 36, 39) intersect with none in the database. These are either reported heatwaves or droughts, and one reported compound event. Focusing on one grid cell of that particular reported compound event in Canada (obs_event 8 in Table A1), Figure 10 shows that, although both dry and hot conditions were observed over its reported course, they did not coincide according to the stringent threshold used in this study. When the strong heatwave started on 25 June 2021, the PEI values were still above the 10th percentile. Later, they dropped under the first percentile, but the maximum daily temperatures, although above the 90th percentile, remained below the 99th percentile except for two consecutive days, without reaching the three consecutive days necessary to qualify as an extremely dry and hot event in the database proposed here. The summer heatwave of 2022 in Tunisia (event 25) did not trigger the detection workflow presented here (Suppl. Fig A3). Although the $PEI_{30}$ did drop below the 1st percentile, the ERA5 maximum temperature did

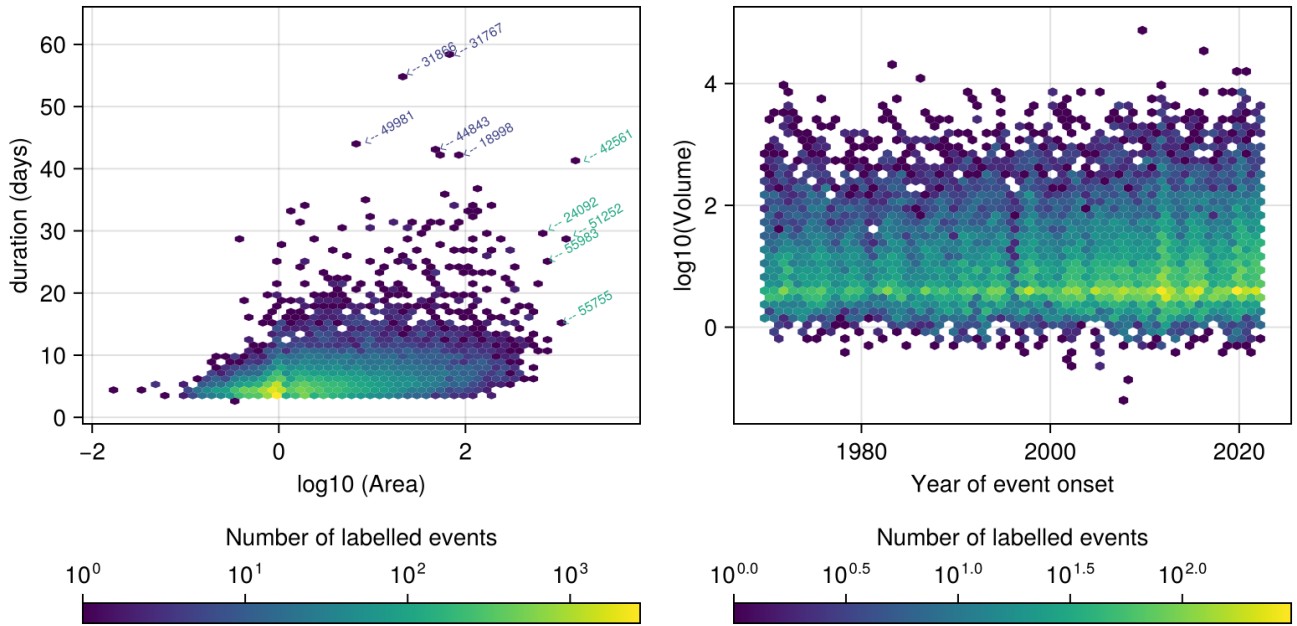

**Figure 7.** Two-dimensional histograms of labelled events over land only in the years 1970 to 2022. Left: Duration versus area of events. Labels indicate the five events with the largest volume (green) and the longest duration (blue); Right Volume of event versus year of event's onset.

not reach the 48 ° C reported in Tunis on 13 July 2022 (Pratt, 2022) and the indicator stayed under the 99th percentile. The Southern Great Plains drought of 2006 (event 31) occurred in winter and was hence not associated with a heatwave as defined in this study. The 1993 drought in North East Brazil (event 36) and the Sahel drought (event 39) of 1983-1984 were not associated

with heatwaves, although extreme heat and drought coincided in the previous hydrological year in the Sahel (reported event 40).

## 4   Discussion and outlook

The global event detection of compound dry and hot extreme events faces the difficulty of dealing with processes that happen at different time scales. Droughts occur over months and years while heatwaves take place over a few days or weeks. Computing a

230 global standardized drought index based on daily data proved difficult. Instead, we rely on the empirical probability distribution of the drought and heat indicators. The local rank-transformation assumes that the number of extremes is the same in each grid cell, defined by a global probability. Finding a good threshold for defining extreme events on daily data was also challenging. Selecting DEOs that are too frequent leads to connectivity issues and very large labelled events spanning over the whole globe. Part of the problem was that the connected component analysis for the event detection is run on an equi-rectangular



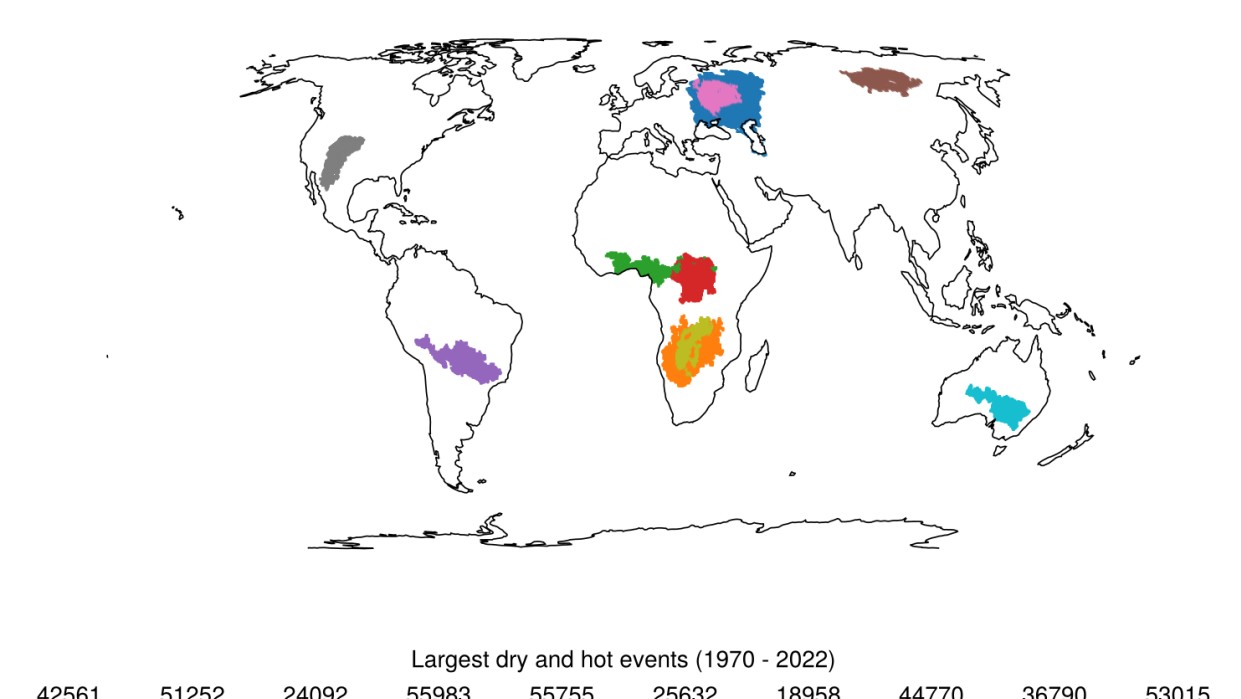

**Figure 8.** Spatial footprint of the ten largest labelled dry and hot events detected in this study from 1970 to 2022 over land.

grid, which leads to a bias towards more connections and larger events in high latitudes. We tested different thresholds and spatial filtering of extreme event scores respecting the spherical nature of the Earth to find a balance between the detection of documented events and avoiding too large events. Other authors have reported similar difficulties when tuning a clustering algorithm to build a database of drought events (Cammalleri et al., 2023). The final threshold is a compromise between the `volume`, `duration` and spatial footprint of the largest labelled extreme events and the effective detection of reported extreme

events. We prefer smaller events to very large ones, even if a reported event is then associated with multiple smaller labelled events from our database. The framework presented here concentrates on detecting and labelling droughts and heatwaves and their compound occurrence based on daily meteorological data. The resulting labelled events can be used to analyze trends at regional, continental and global scales and to drive further research into the impacts of such events on ecosystems, specific species or society. For example, it has been firstly used as a basis for sampling high-resolution satellite imagery (Ji et al., 2024)

to investigate how these compound dry and hot extreme events impact the performance of models predicting the ecosystem state. In addition, the combination of the atmospheric extreme event database and the satellite imagery describing the ecosystem

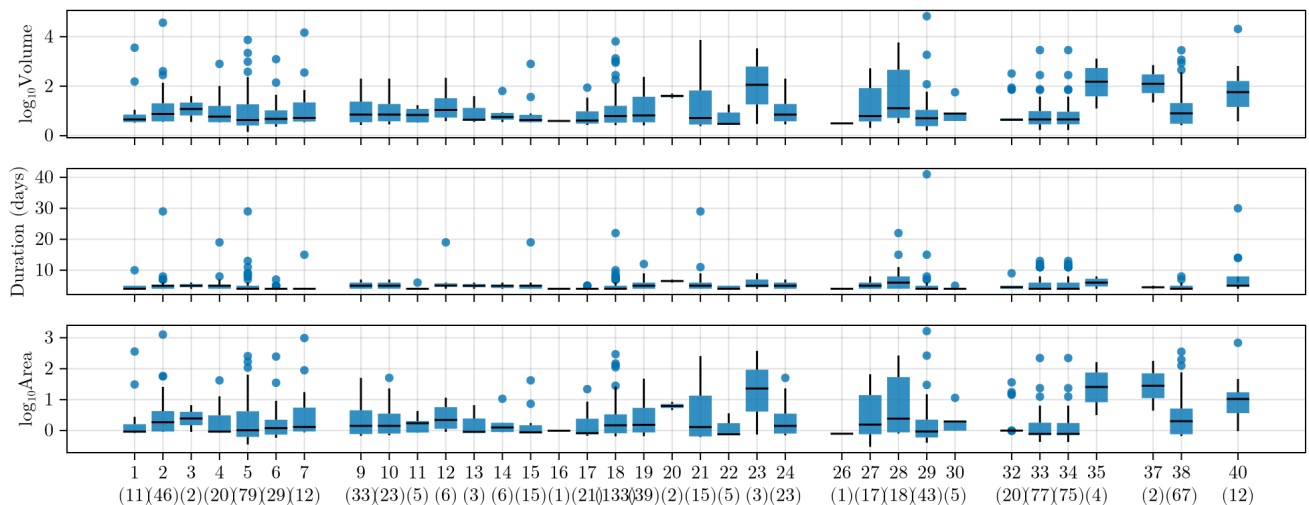

**Figure 9.** Validation of database. `Volume`, `Duration`, `Area` and number (between brackets) of labelled events intersecting the spatio-temporal footprint of events reported in Table A1. An empty space is left for reported events with no intersecting labelled events.

responses can help to improve our understanding of the conditions under which a certain atmospheric extreme event will have impacts on the biosphere.

In the present case, the database includes only dry and hot compound events. However, the event detection pipeline is set
up to be used in a generic way and could produce event databases for different sorts of events. For example, it would be interesting to investigate other types of meteorological extreme events, e.g. involving heavy precipitation, storms, extreme cold and their combinations with heatwaves and droughts. These databases could then be used on their own or for determining areas of interest where they can be combined with other data streams, e.g. to study time series of high-resolution satellite imagery. In addition to the potential of investigating other event types, methodological improvements to the event detection pipeline
are envisioned in future research. The connectivity problem at high latitudes can be addressed using other spatial filtering of extreme event scores respecting the spherical nature of the Earth, or even running the detection pipeline on grid systems with less distortion (DGGS). Besides, the current workflow is based on univariate distributions of indicators of extreme conditions. The compound nature of multi-hazard extreme events could be better apprehended with multivariate distributions. For example, standard multivariate normal kernel has been shown to outperform univariate extreme event detection on synthetic data (Flach
et al., 2017) and successfully applied on real Earth system data to detect anomalies (Flach et al., 2021). Moreover, the addition of new data to the database currently necessitates to run again the complete workflow to update the rank transformed indicators. It also bears the risk that previously detected extremes don't appear as extremes if there is a distribution shift, which seems to be the case as shown in Figure 5. Besides, the labels would not be consistent across versions. Hence, in future versions, we will determine the thresholds based on a reference period, which will facilitate the addition of updated data and will ensure
that previously detected extreme events stay valid. In its current state, the database records the extreme events, but not their





**Figure 10.** Reported compound dry and hot extreme event in British Colombia not associated with labelled event from the proposed database. Although dry and hot conditions were observed over its reported course, they did not coincide according to the stringent threshold used in this study. Panels show (a) the maximum daily temperature, (b) the daily precipitation and reference evapotranspiration, (c) the three drought indicators (PEI) and (d) the Discrete Extreme Occurrences (DEO).



intensity. A combined cumulative metric for both dry and hot conditions would need particular attention. In their review, Hao et al. (2022) mention the Dry-Hot Magnitude Index (DHMI) of compound dry and hot extremes (Wu et al., 2019) valid for monthly input data. It could be adapted to deal with the daily data and with the multiple drought indicators used in Dheed.

## 5 Conclusions

In this data description paper, we propose Dheed, a daily dry and hot extreme events database based on ERA5 consisting of two data cubes and a table: (i) an EventCube of Discrete Extreme Occurrences (DEOs), i.e. days in which extremely dry and/or hot conditions were detected; (ii) a LabelCube of uniquely labelled Dry and Hot Extreme Events (DHEEs), i.e. blobs of simultaneously dry and hot DEOs connected in space and time; (iii) StatEvents, a table containing summary statistics for all labelled DHEEs. The analysis of the EventCube confirms that the occurrence of both dry and hot extremes as well as their
co-occurrence has increased significantly in the past few decades. The trend is not homogeneous across all continents, with Europe and Africa seeing the strongest increase in the annual number of days and areas affected by DHEEs. The largest and longest DHEEs were tracked in the scientific literature and the Dheed was compared against a list of extreme events reported in the literature and collected a priori. The labelCube and its associated table allow the user to easily retrieve in time and space extremely dry and hot conditions, which have occurred, according to climate reanalysis data, between 1950 and 2022, to
further study their impact on ecosystems and societies.

## 6 Code and data availability

Code associated with this study, including the full data processing to create the database of dry and hot extreme events, as well as the creation of the figures presented in this article, is available from zenodo/10.5281/zenodo.13711289 (Weynants et al., 2024a). The database of connected compound dry and hot extreme events is available from zenodo/10.5281/zenodo.11546130
(Weynants et al., 2024b). With no guarantee of permanent storage, all data cubes generated with the current workflow can currently be accessed on a public s3 bucket at https://s3.bgc-jena.mpg.de:9000/minio/deepextremes/v3/. A ReadMe file details the contents of the data store and how to access the data cubes with Julia or python: https://s3.bgc-jena.mpg.de:9000/minio/deepextremes/v3/ReadMe.md.

## Appendix A: Supplementary material

### A1 Time series from Dheed

Figure A1 shows the last five years of the timeseries for the four indicators ($T_{2m,max}$, $PEI_{30}$, $PEI_{90}$, and $PEI_{180}$) used in the detection of DEOs around the city of Jena, Germany (50.9° North, 11.59° West). Daily $ET_{ref}$ and $P$ are also shown in the background. At that particular location, the 1% threshold of maximum daily temperature obtained for the full timeseries (1950–2022) is 304.51 K, or 31.36 ° C. Such a threshold classifies as extremes only the summer hot days. The thresholds



for the drought indicators are $PEI_{30} = -1.31$, $PEI_{90} = -0.82$ and $PEI_{180} = -0.43$ mm/day. 2018, 2019, 2020 and 2022 have been dry, with cumulative water deficit showing for all three PEI. At a location in a completely different climate zone, the thresholds will also be different. For example, around Niamey, Niger (13.5116° N, 2.1254° E, Fig A2), the thresholds are: $T_{2m,max} = 44.00$, $PEI_{30} = -5.02$, $PEI_{90} = -4.70$ and $PEI_{180} = -4.31$. In this Sahelian climate, a deficit in water is the norm rather than the exception.





**Figure A1.** Timeseries (2018–2022) of (a) maximum daily temperature, (b) daily precipitation and reference evapotranspiration, (c) the three drought indicators (PEI) and (d) the Discrete Extreme Occurrences (DEO) around the city of Jena, Germany. The summers in those years (except 2021) were relatively dry, with very hot days resulting in compound dry and hot extremes in 2018, 2019 and 2022.







**Figure A2.** Timeseries (1981–1985) of (a) maximum daily temperature, (b) daily precipitation and reference evapotranspiration, (c) the three drought indicators (PEI) and (d) the Discrete Extreme Occurrences (DEO) around the city of Niamey, Niger. The year 1983 was very dry, but it had only one very hot day resulting in a compound dry and hot event that would not be labelled in Dheed, where events must last at least three days.



**Figure A3.** Reported compound dry and hot extreme event in Tunisia not associated with labelled event from the proposed database. Although dry conditions were observed over its reported course, the maximum temperature did not reach the stringent threshold used in this study. Panels show (a) the maximum daily temperature, (b) the daily precipitation and reference evapotranspiration, (c) the three drought indicators (PEI) and (d) the Discrete Extreme Occurrences (DEO).



**A2   Supplementary table**



**Table A1.** Extreme events reported in the literature or the media used to validate the event detection method.

| Event | Region | Type | Start | End | West | East | South | North | Ref. |
|---|---|---|---|---|---|---|---|---|---|
| 1 | South Africa | heatwave | 2016-01-01 | 2016-01-10 | 18.0 | 48.0 | -35.0 | -16.0 | Meque et al. (2022) |
| 2 | South Africa | drought | 2016-10-07 | 2017-01-30 | 18.0 | 48.0 | -35.0 | -16.0 | Meque et al. (2022) |
| 3 | Pakistan | heatwave | 2017-05-20 | 2017-06-02 | 60.5 | 77.25 | 23.5 | 37.25 | Wikipedia (2017) |
| 4 | India-Pakistan | drought | 2019-02-01 | 2019-06-30 | 61.0 | 89.0 | 7.0 | 34.0 | Wikipedia (2019) |
| 5 | Europe | compound | 2018-06-01 | 2018-08-31 | -10.0 | 35.0 | 30.0 | 70.0 | Liu et al. (2020) |
| 6 | Europe | compound | 2019-06-01 | 2019-08-31 | -10.0 | 35.0 | 30.0 | 70.0 | Bastos et al. (2021) |
| 7 | Brazil | compound | 2020-09-20 | 2020-11-10 | -56.5 | -18.5 | -56.5 | -18.5 | Libonati et al. (2022) |
| 8 | Canada | compound | 2021-06-20 | 2021-07-10 | -127.0 | -95.0 | 48.0 | 60.0 | White et al. (2023) |
| 9 | Europe | drought | 2022-03-01 | 2022-07-22 | -10.0 | 37.0 | 30.0 | 54.0 | Tripathy and Mishra (2023) |
| 10 | Europe | heatwave | 2022-07-10 | 2022-7-22 | -10.0 | 37.0 | 30.0 | 54.0 | Tripathy and Mishra (2023) |
| 11 | India-Pakistan | compound | 2022-03-15 | 2022-05-30 | 61.0 | 89.0 | 7.0 | 34.0 | Aadhar and Mishra (2023) |
| 12 | India | heatwave | 2016-04-01 | 2016-05-20 | 61.0 | 89.0 | 7.0 | 34.0 | Singh et al. (2017) |
| 13 | India | heatwave | 2017-04-12 | 2017-06-15 | 61.0 | 89.0 | 7.0 | 34.0 | Hari and Tyagi (2021) |
| 14 | India | heatwave | 2018-05-12 | 2018-06-10 | 61.0 | 89.0 | 7.0 | 34.0 | Safi (2018); Hari and Tyagi (2021) |
| 15 | India | heatwave | 2019-06-01 | 2019-06-30 | 61.0 | 89.0 | 7.0 | 34.0 | Hari and Tyagi (2021) |
| 16 | India | heatwave | 2022-03-01 | 2022-03-31 | 61.0 | 89.0 | 7.0 | 34.0 | Aadhar and Mishra (2023) |
| 17 | USA | drought | 2017-03-01 | 2017-12-31 | -125.0 | -70.0 | 25.0 | 50.0 | NOAA (2018) |
| 18 | USA | drought | 2020-01-01 | 2020-12-31 | -125.0 | -70.0 | 25.0 | 50.0 | NOAA (2021) |
| 19 | USA | drought | 2021-01-01 | 2021-12-31 | -125.0 | -70.0 | 25.0 | 50.0 | NOAA (2022) |
| 20 | W. North America | heatwave | 2021-06-25 | 2021-07-07 | -140.0 | -115.0 | 35.0 | 65.0 | NOAA (2022) |
| 21 | Europe-middle | heatwave | 2018-07-01 | 2018-07-30 | -3.0 | 23.0 | 42.0 | 53.0 | Rousi et al. (2023) |
| 22 | Europe-west | heatwave | 2019-06-24 | 2019-06-30 | -9.0 | 16.0 | 35.0 | 60.0 | Xu et al. (2020) |
| 23 | Europe-midwest | heatwave | 2020-06-01 | 2020-08-16 | -9.0 | 5.0 | 42.0 | 60.0 | Copernicus |
| 24 | Europe | heatwave | 2022-07-10 | 2022-07-25 | -10.0 | 35.0 | 30.0 | 70.0 | Pratt (2022) |
| 25 | Tunisia | heatwave | 2022-07-10 | 2022-07-25 | 7.5 | 12.0 | 30.0 | 38.0 | Pratt (2022) |
| 26 | Iran | heatwave | 2022-07-10 | 2022-07-25 | 44.0 | 63.5 | 24.5 | 40.03 | Pratt (2022) |
| 27 | China | heatwave | 2022-07-10 | 2022-07-25 | 53.5 | 73.5 | 8.5 | 135.0 | Pratt (2022) |
| 28 | Texas, USA | compound | 2011-06-01 | 2011-08-31 | -106.65 | -93.51 | 25.84 | 36.5 | Nielsen-Gammon (2012) |
| 29 | Russia | heatwave | 2010-06-01 | 2010-08-30 | 28.75 | 60.25 | 48.25 | 66.75 | Flach et al. (2018) |
| 30 | Amazon | drought | 2010-01-01 | 2010-12-31 | -73.0 | -64.0 | -11.0 | -4.0 | Lewis et al. (2011) |
| 31 | USA* | drought | 2005-11-01 | 2006-02-28 | -100.0 | -95.0 | 32.5 | 37.5 | Dong et al. (2011) |
| 32 | Amazon | drought | 2005-01-01 | 2005-12-31 | -73.0 | -64.0 | -11.0 | -4.0 | Lewis et al. (2011) |
| 33 | Europe | drought | 2003-07-01 | 2003-09-30 | -10.0 | 35.0 | 35.0 | 65.0 | Ciais et al. (2005) |





| obs_event | Region | Type | Start | End | West | East | South | North | Ref. |
|---|---|---|---|---|---|---|---|---|---|
| | *continued from previous page* | | | | | | | | |
| 34 | Europe | heatwave | 2003-07-01 | 2003-08-31 | -10.0 | 35.0 | 35.0 | 65.0 | Ciais et al. (2005) |
| 35 | North Argentina | drought | 1995-07-01 | 1996-06-30 | -75.0 | -56.0 | -40.0 | -24.0 | Minetti et al. (2003) |
| 36 | North East Brazil | drought | 1993-02-01 | 1993-05-31 | -47.0 | -35.0 | -12.0 | 7.5 | Rao et al. (1995) |
| 37 | Poland | drought | 1992-09-01 | 1992-09-30 | 14.0 | 24.0 | 49.0 | 55.0 | Łabędzki (2007) |
| 38 | USA | drought | 1988-03-01 | 1988-07-31 | -160.0 | -50.0 | 30.0 | 60.0 | Namias (1991) |
| 39 | Sahel | drought | 1983-10-01 | 1984-09-30 | -10.0 | 33.0 | 10.0 | 18.0 | Tucker et al. (1986) |

*Southern Great Plains



*Author contributions.* MW, FG and NL in conversation with MDM and the Deep Extremes team designed the methodology. NL, FG and
305 MW coded the workflow. UW retrieved and pre-processed the ERA5 data. CJ compiled the table of historic extreme events. MW ran the
code, conducted the analyses and wrote the manuscript with contributions from all co-authors.

*Competing interests.* The authors declare that they have no conflict of interest.

*Disclaimer.* The data are provided as is, with no warranties.

*Acknowledgements.* This work was funded by the European Space Agency (ESA) AI4Science project "Multi-Hazards, Compounds and
310 Cascade events: DeepExtremes," 2022–2024, and the European Union's Horizon 2020 research and innovation program within the project
"XAIDA: Extreme Events – Artificial Intelligence for Detection and Attribution", (grant agreement 101003469). Recent developments in the
Julia package YAXArrays.jl were funded by ESA AI4Science project "The DeepESDL AI-Ready Earth System Data Lab".



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
