# Peer review of "Dheed: an ERA5 based global database of compound dry and hot extreme events from 1950 to 2023"

_Earth System Science Data, 2024_

## Referee Comment (RC2)

**Review of: "Dhhed: and ERA5 based global database of dry and hot extreme events from 1950 to 2022"**

December 18, 2024

The paper presents a new dataset of compound dry and hot extreme events for the historical period 1950-2022, based on the ERA5 reanalysis data. The methodology introduced by the authors represents a valuable effort towards a more unified approach for the characterization of dry and hot days over different land regions. This could be valuable for both climatological and impact studies. Unfortunately, before the paper can be considered for publication in the journal, there are a serious of major issues that need to be carefully addressed by the authors:

**1 Major Concerns**

- The analysis is conducted on data with a seasonal cycle. In order to conduct an analysis of extremes at different locations and time of the year, it is usual to remove the seasonal trend from the data when determining their extremeness.

- Very often in literature, as also mentioned by the authors, heatwaves and droughts are calculated with respect to a reference period. The length of the reference period is normally 30 years. This is usually a compromise between having a sample size large enough for the robust computation of high and low percentiles, as well as having a period where trends related to climate change can be neglected. The authors, on the contrary, calculate their threshold for determining extreme events considering, for each grid box, all the days of the years from 1950 to 2022. This could lead to undesirable biases due to strong trends in the considered variables. Possibly acknowledging this, the authors suggest to calculate their threshold for extreme events with respect to a reference period in future work. I think that this is a very important point that could compromise the reliability of the provided dataset and I would suggest the authors to reperform their analysis, properly defining their threshold with respect to a commonly used 30-year period.

- Also, I think that the authors should better motivate the selection of the 1% threshold for defining extremes. Higher percentiles are more sensitive to the size of the sample used for estimating the underlying distribution of the considered data (See Brunner and Voigt 2024 for further clarifications). For this reason, it is usual to characterize heatwaves using thresholds based on the 90th or 95th percentile (Perkins et al. 2012, Russo et al. 2014, Russo et al. 2015). How would your results change for lower thresholds (i.e. 95th or 90th percentile)?

- In general the text should be improved for clarity. In particular, the method section can be sensibly improved: at the moment the employed methodologies do no always result clear.

- There is large confidence in the literature, often based on the same dataset used by the authors in their study (i.e. ERA5), that heatwaves have increased in extent, length, intensity and number during the historical period. I find it alarming that the authors do not find any significant trend in heat days in their dataset. Hence, I would suggest the authors to carefully review their analysis, also taking into consideration my previous comments.

- Also connected to the previous point, the paper proposes a qualitative evaluation of the dataset based on events commonly known from the literature. I think that such an evaluation, at least qualitatively, should be also conducted for the results of the analysis of the trends in the different considered variables.

- In their method for characterizing compound extreme events, the authors use together 3 variables for precipitation and 1 for temperature? Would this not lead to results that are biased towards droughts rather than heatwaves?

- The Labels of all figures should be improved, possibly including more and better details

**2 Specific Comments**

- The Abstract is very generic. It could be extended and should be reconsidered

- In the abstract, please make it clearer that in your method you consider 1 heat extreme indicator and 3 droughts indicators at the moment.

- Also in the abstract, you mention details of the conducted analysis (such as peak over threshold) that are not reported in the methods

- The structure of the introduction is a bit too general and can be improved

- INtro: you can describe more examples on how extremne events affect the ecosystem and society

- l 12-13: "With Earth climate currently changing": revision needed. The Earth climate has been always changing, over million (if not billions) of years. Here, I think that the authors refer to anthropogenic driven climate change. Please reformulate this period.

- l 16-19: Can you provide more examples on the impact of compound heat-hydrological extremes on vegetation as compared to single events?

- l 22: "The cascading process.. also impact society as a whole": could you provide more details why and in which way?

- l 24-25: I think that there are many studies in the literature working towards a more unified definition of droughts and heatwaves that would be worth to mention here, such as for example:

  *Perkins, Sarah E. "A review on the scientific understanding of heatwaves - Their measurement, driving mechanisms, and changes at the global scale." Atmospheric Research 164 (2015): 242-267.*

- Also, please mention that the lack of a unified way of defining heatwaves or drought is also due to the fact that the definition of these events often depends on the purpose of the study, the considered region and time of a year.

- l33: "compartment": use different wording

- l39-40: reformulation needed

- l43: "can propagate into impacts": reformulate

- l46-47: What was the goal of the studies you mention? which events they have considered? Please provide more details

- l 48-50: please reformulate

- l 52-54: "For example, it can serve ... to train models predicting ... ": can you better describe here what you mean by minicubes of high resolution satellite imagery and how Dheed can be useful for the purpose you mention?

- In the introduction, I think that the advantages of a reliable identification of Dheed can be better detailed.

- For your analysis, which period of the year you consider? You consider all seasons together? Please specify in section 2.1.

- Most of the data for the period 1950-2022 are expected to be characterized by a trend, especially for T2M. How do you take this into account for your analyses?

- In section 2.1 you say that you calculate daily mean, min and max temperatures from hourly T2M values. But which of these 3 variables you use in your analysis is not clear.

- I think it would be better to introduce ETref before mentioning it in line 76

- l77-82: Please reformulate, since the information you provide are not clear. Why You have 60x60 points in longitude and latitudwe for each Zarr cube? Is this a personal choice of the author? Please clarify. Please, also be aware that the original horizontal resolution of ERA5 data is not 0.25 degrees lon. Specify that you use a gridded product, provided on a regular grid with a horizontal resolution of 0.25 degrees.

- l 89: it should be $\Theta_2 m$

- In equation 1, how do you calculate the parameter $C_d$? I see that you provide more details at lines 101-102. I think that it would help a better readibility of this part of the text if details on parameter $C_d$ are reported when the parameter is first introduced.

- In Equation 1, from where you derived the value of G? please specify.

- Please provide reference for equation 2

- l 102: what is $10^{-6}$? the value of changes in ETref when using constant values of $C_d$? please specify

- When aggregatinhg the 3 PEIs with Tmax in your dataset, you will be giving more weights to drought events. Why this choice? Wouldn't it be better to consider only one drought indicator together with the heat indicator? for example, for a drought event lasting more than 90 days, this will be counted 3 times according to your definition.

- l 114-115: "Fitting a parametric distribution... proved difficult": why? please provide more details

- l117: Not clear, please reformulate.

- Why you choose the 1 % as a threshold? Higher percentiles are more sensitive to the sample size used for estimating the underlying data distribution. For this reason, it is usual to characterize heatwaves with thresholds based on the 90th or 95th percentile. How would your results change in this case?

- l 122: how do you define a data cube?

- l 125-126: "have uneven values greater than 1": what is the unit you are considering in this case? Which metric are you considering?

- l. 124: you say that you extract DHEEs as labelled groups of dry and hot DEOs. Following your analysis, I would rather reformulate this sentence. In fact I think it would be more correct to say that you extract DHEEs as labelled groups of dry and/or hot DEOs.

- Also check the text for consistency between DHEEs and dhees

- l.124-133: This part of the methods results not clear and needs reformulation. When do you apply the 3-day long condition? You do this at each grid point, before the temporal and spatial aggregation? What do you mean by "group DEOs connecting across the globe along the longitude dimension"? What about latitudes? What do you mean by "possibility to merge labels from contiguous data cubes"? Do you consider the fact that at different latitudes you have different numbers of land points in your final ranking of events?

- Fig. 2: It would be nice to also see some examples of events occurring over Asia and North America. In the caption, please be consistent in the style of the enumeration of the rows (row 1 and 2 vs Third). Also in the caption, what do you show in row 4? please specify

- l139: what is the percentage of events affected by a single indicator? Please clarify

- l. 153: You go from introducing Fig. 2 to Fig. 10. Please reconsider the order of the figures so that they can be referenced in the same order they appear in the text

- l157-159: not clear. Please reformulate

- l. 162-163: Why not excluding then the years from 1950 to 1970 already before-hand, since you know that for these years ERA5 is less reliable?

- l 165: "but there seems to be a positive trend": in which feature?

- l. 165: It is not clear how you define the different categories of Fig. 3

- Fig. 3: some colors miss a label. What do they indicate?

- l. 160: add some references on which your considerations are based

- how the 4th row of Fig. 3 differs from the plot of Fig. 5? pleae clarify

- l. 174: can you better clarify what you mean by volume?

- Caption of Fig 7: please specify that in this case you are only considering hot and dry events

- Table 2: Maybe I would join Fig. 8 and Table 2. This would help readability.

- section 3.3: A similar validation based on available literature should be conducted also for the evinced trends.

- l 213: which database?

- l.214: Provide more context on the specific events, as well as references

- Fig. 8: Maybe you can label the events by year? what are the two events occurring in Russia?

- l 238: "the final threshold": What is this final threshold?

- l. 235-240: Can you better report about this in the methods section?

---

## Author Comment (AC1)

*We thank the reviewer for their comments and suggestions. Please find hereafter our reply in italics shaded in blue.*

**RC2**: 'Comment on essd-2024-396', Anonymous Referee #2, 18 Dec 2024 reply

The comment was uploaded in the form of a supplement: https://essd.copernicus.org/preprints/essd-2024-396/essd-2024-396-RC2-supplement.pdf

**Citation**: https://doi.org/10.5194/essd-2024-396-RC2

The paper presents a new dataset of compound dry and hot extreme events for the historical period 1950-2022, based on the ERA5 reanalysis data. The methodology introduced by the authors represents a valuable effort towards a more unified approach for the characterization of dry and hot days over different land regions. This could be valuable for both climatological and impact studies. Unfortunately, before the paper can be considered for publication in the journal, there are a serious of major issues that need to be carefully addressed by the authors:

**1 Major Concerns**

• The analysis is conducted on data with a seasonal cycle. In order to conduct an analysis of extremes at different locations and time of the year, it is usual to remove the seasonal trend from the data when determining their extremeness.

*We thank the reviewer for highlighting this. We are well aware that it is usual to deseasonalize the data before detecting the extremes. However, we decided to use the natural values with absolute local thresholds rather than calculating thresholds on the anomalies to the mean seasonal cycle. The rationale for our choice is twofold. First, in a changing climate, seasons are shifting and analysing extremes on the anomalies may introduce biases. Second, the primary purpose of this database was to explore the impacts of the compound extremes on the vegetation. There is a growing literature confirming that the physiological impacts of extreme temperatures and dryness are more absolute than relative.*

• Very often in literature, as also mentioned by the authors, heatwaves and droughts are calculated with respect to a reference period. The length of the reference period is normally 30 years. This is usually a compromise between having a sample size large enough for the robust computation of high and low percentiles, as well as having a period where trends related to climate change can be neglected. The authors, on the contrary, calculate their threshold for determining extreme events considering, for each grid box, all the days of the years from 1950 to 2022. This could lead to un-desirable biases due to strong trends in the considered variables. Possibly acknowledging this, the authors suggest to calculate their threshold for extreme events with respect to a reference period in future work. I think that this is a very important point that could compromise the reliability of the provided dataset and I would suggest the authors to reperform their analysis, properly defining their threshold with respect to a commonly used 30-year period

*In a changing climate, restricting the reference period to 30 years for a multi-decadal analysis could introduce biases (Sippel et al. 2015, doi/10.1002/2015GL066307), especially in presence of*

*non-stationarity in the time series. We prefer analysing the longest time series possible to derive the experimental distributions of the variables of interest. Future updates of the database will however rely on the threshold determined on the current version so as to not modify the existing labels. Hence, the associated probability won't be the nominal one for all years after the reference period.*

• Also, I think that the authors should better motivate the selection of the 1% threshold for defining extremes. Higher percentiles are more sensitive to the size of the sample used for estimating the underlying distribution of the considered data (See Brunner and Voigt 2024 for further clarifications). For this reason, it is usual to characterize heatwaves using thresholds based on the 90th or 95th percentile (Perkins et al. 2012, Russo et al. 2014, Russo et al. 2015). How would your results change for lower thresholds (i.e. 95th or 90th percentile)?

*The 1% threshold was chosen after testing different thresholds (10%, 5%, 2.5%, 1%, 0.5%). We ran the analysis globally, also over the ocean to allow for extreme conditions prevailing in land masses separated by some water bodies to be associated with the same labelled event. However, larger thresholds led to connected compound events that were spanning the whole globe and/or lasting more than a year. Therefore, we adopted the largest threshold that was creating events of reasonable size.*

• In general the text should be improved for clarity. In particular, the method section can be sensibly improved: at the moment the employed methodologies do no always result clear.

*We thank the reviewer for this remark. We have modified the text of the method section in a revised manuscript. Hopefully it is now clearer for the reader.*

• There is large confidence in the literature, often based on the same dataset used by the authors in their study (i.e. ERA5), that heatwaves have increased in extent, length, intensity and number during the historical period. I find it alarming that the authors do not find any significant trend in heat days in their dataset. Hence, I would suggest the authors to carefully review their analysis, also taking into consideration my previous comments.

*We thank the reviewers for bringing our attention to this lack of consistency with the literature, which also puzzled us. After a careful review of the code, we identified an error in our processing of the original ERA5 data, which introduced a monthly bias in the temperature and precipitation data. We have run the workflow again after correction and the results are now consistent with the literature, showing a significant positive trend in the global number of extremely hot days.*

• Also connected to the previous point, the paper proposes a qualitative evaluation of the dataset based on events commonly known from the literature. I think that such an evaluation, at least qualitatively, should be also conducted for the results of the analysis of the trends in the different considered variables.

*We thank the reviewer for this suggestion. We have added the following comparative statements regarding the trends:*

*The results of the trend analysis presented in the previous section are consistent with the literature even if no other study relies on the exact same definition of CDH as the one we use here. Using three*

*different combinations of observed and reanalysis-based data sets, \citet{mukherjee_increase_2021} note a significant increase in global drought-related heat waves and their corresponding spatial extent in a recent (warmer) period (2000--2016) compared to a past period (1983--1999). Combining forecasting and reanalysis data and a ten-year return period, \citet{zampieri_stationarity_2024} also observe a significant increase in area subject to drought (0.5\% of land area per decade), heat risk (7.3\% in recent decades) and their compound (about 0.6\% per decade) over the period 1983--2023 (reference period 1993–2016). They observe similar albeit less pronounced results with stationary thresholds and time-dependant percentiles or thresholds.*

• In their method for characterizing compound extreme events, the authors use together 3 variables for precipitation and 1 for temperature? Would this not lead to results that are biased towards droughts rather than heat- waves?

*We thank the reviewer for bringing our attention to this potential bias. We want to stress that the labelled events are always extremely hot and that these hot conditions need to last at least three consecutive days. The rationale that motivated the choice of combining the heat indicator with three drought indicators is the production of a non overlapping database of compound events. We could also produce separate databases for the different accumulation periods of water stress, however the entries could overlap. Therefore, we produced a single database in which the user can retrieve the proportion of the event incurred to different accumulation periods. For example, a user interested in short droughts excluding longer ones would select entries with a large coverage of PEI_30 and small coverage of PEI_90 and PEI_180. This allows for a finer use of the database.*

• The Labels of all figures should be improved, possibly including more and better details

*The figure captions will be improved in a revised manuscript.*

**2 Specific Comments**

• The Abstract is very generic. It could be extended and should be reconsidered

• In the abstract, please make it clearer that in your method you consider 1 heat extreme indicator and 3 droughts indicators at the moment.

• Also in the abstract, you mention details of the conducted analysis (such as peak over threshold) that are not reported in the methods

*The abstract has been re-written as follows:*

*The intensification of climate extremes is one of the most immediate effects of global climate change. Heatwaves and droughts have uneven impacts on ecosystems that can be exacerbated in case of compound events. To comprehensively study these events, e.g. with local high-resolution remote sensing or in-situ data, a global catalogue of compound dry and hot (CDH) events is essential. Here, we propose a database of large-scale dry and hot extreme events based on ERA5 climate reanalyis data. Drought indicators are constructed based on the balance between reference evapotranspiration and precipitation averaged over 30, 90 and 180 days. Extreme events are detected with absolute local thresholds for the 1950–2023 period. Extremes are defined as daily maximum temperature at 2*

*m exceeding a 99% threshold based on the experimental probability distribution, combined with any of the three drought indicators falling short of the 1% threshold. Unique labels are assigned to CDH events lasting at least three days using a connected component analysis. Their spatiotemporal extent and summary statistics are extracted for all labelled events. The identified CDH events are validated against extreme events reported in the literature. Out of 40 events listed a priori, 38 could be associated with labelled CDH events. All 10 largest and 10 longest labelled CDH events could be linked to droughts and/or heatwaves reported in the scientific or grey literature. The Dheed database of connected compound dry and hot extreme events is available from zenodo/10.5281/zenodo.11546130 (Weynants et al., 2025).*

• The structure of the introduction is a bit too general and can be improved

• INtro: you can describe more examples on how extremne events affect the ecosystem and society

*The following sentence has been added: Increased heat and drought stress on vegetation challenges the role of ecosystems as carbon sinks, e. g. through contributing to altered primary productivity (Bastos et al.), increases in forest mortality (Senf et al.), risk of intensifying wildfires (Cunningham et al.; Jain et al.), and long-lasting impacts on above-ground biomass (Yang et al.).*

• l 12-13: "With Earth climate currently changing": revision needed. The Earth climate has been always changing, over million (if not billions) of years. Here, I think that the authors refer to anthropogenic driven climate change. Please reformulate this period.

*The introductory sentence has been reformulated:*

*With the current anthropogenic-driven climate change, the intensity and frequency of heat and hydroclimatic extremes are increasing (Seneviratne et al., 2023; Rodell and Li, 2023).*

• l 16-19: Can you provide more examples on the impact of compound heat-hydrological extremes on vegetation as compared to single events?

*The following sentence has been added:*

*Strong negative impacts of concurrent heat and drought as compared to univariate extremes are also evident in agricultural losses, e. g. in soybean yields (Hamed et al., 2021).*

• l 22: "The cascading process.. also impact society as a whole": could you provide more details why and in which way?

*The sentence has been modified in the revised ms:*

*The cascading processes triggered by concurrent DH extremes also impact society as a whole (Niggli et al., 2022), and require particular focus given the expected increasing burden on society by DH in many parts of the world under anthropogenic climate change (Zhang et al., 2024; Ridder et al., 2022). For example, concurrent DH extremes are projected to impact global food security (Biess et al., 2024; Kornhuber et al., 2020). Global, open data on DH events thus also forms an important basis in providing information for guiding policy decisions (Raymond et al., 2020)*

• l 24-25: I think that there are many studies in the literature working towards a more unified definition of droughts and heatwaves that would be worth to mention here, such as for example: Perkins, Sarah E. "A review on the scientific understanding of heatwaves - Their measurement, driving mechanisms, and changes at the global scale." Atmospheric Research 164 (2015): 242-267.

*Reference to the Perkins review has been added as:*

*\citep{perkins_review_2015} recognise the difficulty to settle on a universal definition of heatwaves that fit all sectors, but also highlight the need to reduce the large number of metrics currently used.*

• Also, please mention that the lack of a unified way of defining heatwaves or drought is also due to the fact that the definition of these events often depends on the purpose of the study, the considered region and time of a year.

*We thank the reviewer for this suggestion. This was already stated at line 24, but a clearer statement has now been added, using their own words:*

*[Yet, the definition of heatwaves and droughts is not standardized in the literature], often depending on the purposes of the study, the considered region and the time of the year}.*

• l33: "compartment": use different wording

*The sentence has been reformulated as:*

*Droughts are prolonged dry periods that can last from weeks to years. Their typology depends on their duration and intensity, with diverse impacts on ecosystems. One generally distinguishes between meteorological, hydrological, agricultural and socio-economic droughts (Mishra and Singh, 2010).*

• l39-40: reformulation needed

*The sentence "The rationale is that, otherwise, e.g. a four week drought happening across two months might remain undetected in monthly data." has been reformulated as:*

*Indeed, a short drought, e.g. a four week drought, happening across two months might remain undetected in monthly data.*

• l43: "can propagate into impacts": reformulate

*The sentence has been reformulated as:*

*[...] can cause substantial stress to vegetation and ecosystems in general [...]*

• l46-47: What was the goal of the studies you mention? which events they have considered? Please provide more details

*The sentence has been reformulated as:*

*Studies on the impacts of drought and heat on the biosphere, primary productivity or ecosystems have often focused on single compound events \citep[e.g.,][]{flach_contrasting_2018, ciais_europe-wide_2005, bastos_direct_2020}. Daily drought indices have been computed for specific regions or measurement stations \citep[e.g.,][]{li_standardized_2021, pohl_long-term_2023}, but to the best of our knowledge no global gridded analysis of CDH events at daily scale has been published so far.*

• l 48-50: please reformulate

*The sentence has been reformulated as:*

*In this study we introduce Dheed, a global database of large-scale dry and hot extreme events, product of an extensive analysis of long-term ERA5 global climate reanalysis data \citep{hersbach_era5_2020,hersbach_era5_2023} provided by the European Centre for Medium Range Weather Forecasts (ECMWF).*

• l 52-54: "For example, it can serve ... to train models predicting ... ": can you better describe here what you mean by minicubes of high resolu- tion satellite imagery and how Dheed can be useful for the purpose you mention?

*Minicubes are small data cubes. In Ji et al. (2025), a previous version of Dheed was used to create DeepExtremesCubes, a dataset of The sentence has been reformulated as:*

*For example, it can guide the sampling of small data cubes of high-resolution satellite imagery -- e.g., Copernicus Sentinel-2 data (Ji, Fincke et al. 2025) -- to train models predicting ecosystem states () under extreme climate conditions.*

• In the introduction, I think that the advantages of a reliable identification of Dheed can be better detailed.

*The following paragraph has been added at the end of the introduction.*

*A reliable spatiotemporal identification of past CDH events offers several advantages. (i) Understanding the historical patterns and frequency of these events can help in assessing the risk and potential impact on ecosystems, water resources, and human health. (ii) Policymakers can use this information to develop strategies for mitigation and adaptation, such as water management plans and heat action plans. (iii) Identifying regions most affected by these events allows for targeted allocation of resources and emergency services. (iv) Educating the public about the likelihood and potential impact of these events can enhance community preparedness and resilience. (v) This study provides a valuable dataset for researchers studying climate change and its impacts on extreme weather patterns. Overall, better identification helps in building resilience against future climate extremes.*

• For your analysis, which period of the year you consider? You consider all seasons together? Please specify in section 2.1.

*An absolute threshold is considered at each location. The first sentence of section 2.2 (Event detection) has been reformulated:*

*[The detection of DEOs is based on a purely probabilistic threshold applied to the empirical distribution of the indicators considering the full time series at each location, without removing the mean seasonal cycle, nor any trend.*

• Most of the data for the period 1950-2022 are expected to be characterized by a trend, especially for T2M. How do you take this into account for your analyses?

*The trend expected in the time series is not dealt with specifically, except in the fact that the whole time series is considered to derive the experimental probability distribution. The introductory paragraph to section 2.2 Event detection has been rewritten in the revised manuscript, highlighting the particularities of the detection method and their justification.*

*The detection of DEOs is based on a purely probabilistic threshold applied to the empirical distribution of the indicators, considering the full time series at each location, without removing the mean seasonal cycle, nor any trend. We use an absolute threshold specific to each spatial grid cell to focus on extreme hot and extreme dry conditions, and do not consider here winter warm spells nor relative droughts. The rationale behind this choice is twofold. First, in a fast changing climate, seasons are shifting and analysing extremes on the anomalies may introduce biases. Second, the primary purpose of this database is to explore the impacts of the compound extremes on the vegetation. There is a growing literature confirming that the physiological impacts of extreme temperatures and dryness are more absolute than relative (e.g. Marchin et al. 2022)*

• In section 2.1 you say that you calculate daily mean, min and max temperatures from hourly T2M values. But which of these 3 variables you use in your analysis is not clear.

*Only Tmax is used to detect the extremely hot days, as is explained in section 2.2 (Event detection). Tmin and Tmean are only used as additional information in the database. We added the following sentence at the end of section 2.1:*

*The daily maximum temperature (Tmax) is used as heatwave indicator.*

• I think it would be better to introduce ETref before mentioning it in line 76

*Yes, thank you for the suggestion. The revised manuscript now reads: "and the reference evapotranspiration ET0 (see hereafter)."*

• l77-82: Please reformulate, since the information you provide are not clear. Why You have 60x60 points in longitude and latitudwe for each Zarr cube? Is this a personal choice of the author? Please clarify. Please, also be aware that the original horizontal resolution of ERA5 data is not 0.25 degrees lon. Specify that you use a gridded product, provided on a regular grid with a horizontal resolution of 0.25 degrees.

*The chunking of the Zarr data cube is only mentioned for completeness and to point the user towards the ease of spatio-temporal analyses on the cube, given the chosen chunking. The Zarr data cube is a collection of small compressed files, each holding 60x60x5844 data points.*

*Section 2.1 already started (l70) with the sentence: "The workflow exploits the hourly gridded ERA5 data, from 1950 to 2022", extended to 2023 in the revised version. Nevertheless, in the revised manuscript we mention again that we use the original gridded ERA5 data.*

• l 89: it should be $\Theta 2m$

*Yes, indeed, thank you. The change has been made in the revised manuscript.*

• In equation 1, how do you calculate the parameter Cd? I see that you provide more details at lines 101-102. I think that it would help a better readibility of this part of the text if details on parameter Cd are reported when the parameter is first introduced.

*The sentence relative to Cd has been moved up, where the parameter is first mentioned.*

• In Equation 1, from where you derived the value of G? please specify.

*The value of G is explained at l88-89 in the submitted manuscript.*

• Please provide reference for equation 2

*Reference to Eq 47 in Allen et 1998 has been added in the revised manuscript.*

• l 102: what is 10−6? the value of changes in ETref when using constant values of Cd? please specify

*Yes, the value of change in Etref when using a constant value for Cd. Units (mm/d) have been added for clarity in the revised manuscript.*

• When aggregatinhg the 3 PEIs with Tmax in your dataset, you will be giving more weights to drought events. Why this choice? Wouldn't it be better to consider only one drought indicator together with the heat indicator? for example, for a drought event lasting more than 90 days, this will be counted 3 times according to your definition.

*See reply to previous major concern. The following sentence has been added towards the end of section 2.2:*

*It is worth noting that, given the criteria chosen for the connected component analysis, labelled events are always extremely hot (heat = 100 %) and have a minimum duration of three days. Users can retrieve the proportion of a labelled event incurred to the different drought indicator. For example, a user interested in short droughts excluding longer ones would select entries with a large coverage of PEI_30 and small coverage of PEI_90 and PEI_180. This allows for a finer use of the database respective on the accumulation period.*

• l 114-115: "Fitting a parametric distribution... proved difficult": why? please provide more details

*The sentence has been reformulated as:*

*It is a common procedure to fit a parametric distribution to the PEI data to generate a standardised index (SPEI) with values comparable across space and time. However, the identification of extreme events is based on quantiles only and quantiles can be reliably estimated directly on the data, so we decided to omit the parameter estimation and estimated thresholds based on empirical quantiles directly.*

- l117: Not clear, please reformulate.

*The sentence has been reformulated as:*
*We applied the same rank-transformation to $-T_{max}$. This means that values of $T_{max}$ larger than the 99% quantile will have corresponding values <0.01 in the rank-transformed data. Heatwaves as well as drought events are therefore characterized by low values in their corresponding rank-transformed indicators.*

- Why you choose the 1 % as a threshold? Higher percentiles are more sen- sitive to the sample size used for estimating the underlying data distribu- tion. For this reason, it is usual to characterize heatwaves with thresholds based on the 90th or 95th percentile. How would your results change in this case?

*See reply to previous major concern. The sentence in section 2.2 has been reformulated in the revised manuscript :*

*Different local percentile-based thresholds were tested for detecting extreme conditions (lowest 10 %, 5 %, 2.5 %, 1 %, 0.5 % of the empirical cumulative distributions).*

*and further explanation is given in the Results (3.1):*

*Different local percentile-based thresholds were tested for detecting extreme conditions (not shown). Larger thresholds led to connected compound events that were spanning the whole globe and/or lasting more than a year. Therefore, we adopted the largest threshold that was creating blobs of reasonable size. We chose the lowest 1% as a compromise between the number of data points and the size of the spatio-temporally connected events*

- l 122: how do you define a data cube?

*A data cube is a multidimensional analysis-ready data structure, here with dimensions longitude, latitude, time and variables. A definition has been added in the revised manuscript, at the first instance of the word.*

- l 125-126: "have uneven values greater than 1": what is the unit you are considering in this case? Which metric are you considering?

*The value of 1 refers to the data encoded in the EventCube, where heatwaves are encoded as 0x01, i.e. on the first bit of the byte integer. The sentence has been reformulated as:*

*We restrict the connected component analysis to spatio-temporal grid cells of the EventCube that are both hot (0000000012) and dry (0000000102 OR 0000001002 OR 000010002), i.e. have uneven values greater than 1, if expressed in base 10.*

• l. 124: you say that you extract DHEEs as labelled groups of dry and hot DEOs. Following your analysis, I would rather reformulate this sentence. In fact I think it would be more correct to say that you extract DHEEs as labelled groups of dry and/or hot DEOs.

*No, the DHEEs are 100% hot (uneven values of EventCube) and have at least one drought indicator (see previous comment).*

• Also check the text for consistency between DHEEs and dhees

*Thank you for pointing this out. Corrections have been made in the revised manuscript. For the sake of clarity, a clear distinction is made between compound dry and hot (CDH) events and the dry and hot extreme event database (Dheed).*

• l.124-133: This part of the methods results not clear and needs reformu- lation. When do you apply the 3-day long condition? You do this at each grid point, before the temporal and spatial aggregation? What do you mean by "group DEOs connecting across the globe along the longitude dimension"? What about latitudes? What do you mean by "possibility to merge labels from contiguous data cubes"? Do you consider the fact that at different latitudes you have different numbers of land points in your final ranking of events?

*Thank you for pointing out the lack of clarity in this paragraph. The paragraph has been reformulated as:*

*Moreover, using ImageFiltering.jl (2023) on the time dimension, we filter for events that last at least three consecutive days. The connected component labelling algorithm assigns a unique label to each group of connected DEOs, looking for six way connections. Each grid cell with coordinates (x ± 1, y, z), (x, y ± 1, z) or (x, y, z ± 1) is connected to the grid cell at (x, y, z), with x, y and z the longitude, latitude and time, respectively. We modify the ImageMorphology.label_components function (ImageMorphology.jl, 2023) to group DEOs connecting across the globe along the longitude dimension, allowing for events to connect across the grid longitudinal edge, between 0 and 360 degrees. The connection at high latitudes across the poles is not specifically guaranteed.*

• Fig. 2: It would be nice to also see some examples of events occurring over Asia and North America. In the caption, please be consistent in the style of the enumeration of the rows (row 1 and 2 vs Third). Also in the caption, what do you show in row 4? please specify

*The caption of Figure 2 has been reformulated as:*

*Example of dry and hot extreme event detection workflow over the 2003 summer heatwave in Europe. Columns show the time evolution of the data sampled at every 4th time step from Aug 2 to Aug 14 2003. Rows 1 and 2 show the raw daily maximum 2m air temperature and P EI30 with isolines linking the ranked values at 1%, 10% and 90%. Row 3 shows the encoding into the EventCube where single voxels can be marked as only extremely dry, only extremely hot, a combination of both or none*

*of them.Voxels shown in grey are in a regime of normal conditions. Those shown in white are are in the tails of the distributions, with values smaller than the 10th or greater than the 90th percentile. Row 4 shows the labelled events obtained from the spatio-temporal connected component analysis on the Event-Cube. Only voxels that are both dry and hot, and are connected, are registered with a unique label in the Dheed database of dry and hot extreme events.*

• l139: what is the percentage of events affected by a single indicator? Please clarify

*Each labelled event is 100% extremely hot and extremely dry, but not necessarily all three drought indicators cover the whole event. For example, for event 83007 (2010 Russian drought-heatwave), the 30 days drought indicator is below the 1% threshold for 96.32% of its "volume". The coverages of the 90 and 180 days drought indicators are 59.77% and 28.56% respectively. This particular event is hence mainly concerned by a short term drought. The sentence has been reformulated as:*

*percentage of the event for which each indicator is below the extreme threshold*

• l. 153: You go from introducing Fig. 2 to Fig. 10. Please reconsider the order of the figures so that they can be referenced in the same order they appear in the text

*Figure 10 has been moved up.*

• l157-159: not clear. Please reformulate

*The y axis represents the average percentage of land area affected by an event of a certain type at a given time.*

• l. 162-163: Why not excluding then the years from 1950 to 1970 already before-hand, since you know that for these years ERA5 is less reliable?

*Thank you for this suggestion. We decided to keep the events from 1950 to 1970 in the database so that users could still use those in their analyses. However, we chose to discard them from the trend analysis because our confidence in them is less strong. The sentence has been reformulated as:*

*Therefore, we do not include the years 1950–1969 in the trend analysis. Nevertheless, the Dheed database contains the labelled events from those years.*

• l 165: "but there seems to be a positive trend": in which feature?

*The following sentence has been completed:*

*[The inter-annual variability is large, but there seems to be a positive trend] in the global annual number of extreme dry or hot days. The trends can be further analysed by type of event.*

• l. 165: It is not clear how you define the different categories of Fig. 3

• Fig. 3: some colors miss a label. What do they indicate?

*The caption has been completed and now reads:*

*Annual spatiotemporal extent of extreme dry and hot days, by the value of data in EventCube. The sum of Discrete Extreme Occurrences (DEO) of a given value ($00000001_2$ =1 to $00001111_2$=15) weighted by the cosine of the grid cell latitude is divided by the sum of all land voxels in a given year, expressed as percentage. The shades of blue and purple show the accumulation period of the water balance. The darker the shade the longer the accumulation period: a water balance accumulated over 180 days which is below the 1% threshold is rendered in the darkest shade. The 90 day accumulation period is shown in the medium shade. The 30 day accumulation period has the lightest shade.*

• l. 160: add some references on which your considerations are based

*These considerations are mentioned in Hersbach et al. 2020 (The ERA5 global reanalysis). The reference has been added in the revised manuscript.*

• how the 4th row of Fig. 3 differs from the plot of Fig. 5? pleae clarify

*Figure 3 (now Figure 4) shows the values in EventCube (DEO values from $0001_2$ =1 to $1111_2$=15), coloured as shown in the legend. Figure 4 (now Figure 5) shows the single indicators (for example, if a voxel has DEO = 1001, the same voxel will be counted at Row 1 (heat), DEO & $0001_2$ == $0001_2$ (where & is the bitwise AND) and at Row 4 (d180: drought indicator with 180 day accumulation), DEO & $1000_2$ == $1000_2$). Figure 5 (now Figure 6) counts only DEO that are both hot and dry, i.e. (DEO & $0001_2$ == $0001_2$) AND ((DEO & $0010_2$ == $0100_2$ ) OR (DEO & $0100_2$ == $0100_2$) OR (DEO & $1000_2$ == $1000_2$). The text has been improved in the revised manuscript accordingly.*

• l. 174: can you better clarify what you mean by volume?

*The use of volume here is misleading. It has been replaced by "percentage of extremely dry and hot days and land area" in the revised manuscript.*.

• Caption of Fig 7: please specify that in this case you are only considering hot and dry events

*Thank you for pointing this out. The caption has been corrected in the revised manuscript.*

• Table 2: Maybe I would join Fig. 8 and Table 2. This would help read- ability.

*Thank you for this suggestion. We placed the figure closer to the table in the revised manuscript. This will be pointed to the editor for the potential final composition.*.

• section 3.3: A similar validation based on available literature should be conducted also for the evinced trends.

*Thank you for this suggestion. No trend has been evinced, but the observed trends will be substantiated by references to the literature in the revised manuscript.*
*The results of the trend analysis presented in the previous section are consistent with the literature even if no other study relies on the exact same definition of CDH as the one we use here. Using three different combinations of observed and reanalysis-based data sets, (Mukherjee et al. 2021, DOI:10.1029/2020GL090617) note a significant increase in global drought-related heat waves and*

*their corresponding spatial extent in a recent (warmer) period (2000-2016) compared to a past period (1983-1999). Combining forecasting and reanalysis data and a ten-year return period, Zampieri et al. 2024, DOI:10.1029/2024GL111117) also observe a significant increase in area subject to drought (0.5% of land area per decade), heat risk (7.3% in recent decades) and their compound (about 0.6% per decade) over the period 1983-2023 (reference period 1993–2016). They observe similar albeit less pronounced results with stationary thresholds and time-dependant percentiles or thresholds.*

• l 213: which database?

*the Dheed database (this study).*

• l.214: Provide more context on the specific events, as well as references

*With the corrected workflow, only two reported events from Table 2 do not intersect with any labelled event from Dheed. References are in Table A1*

• Fig. 8: Maybe you can label the events by year? what are the two events occurring in Russia?

*Thank you for this suggestion. The year in which the event started has been added to the figure legend.*

• l 238: "the final threshold": What is this final threshold?

*Different local percentile-based thresholds were tested for detecting extreme conditions (lowest 10 %, 5 %, 2.5 %, 1 %, 0.5 % of the empirical cumulative distributions). The final threshold is 1%, which we used to construct the Dheed. The sentence has been reformulated in the revised manuscript:*

*The 1% threshold is a compromise between the volume, duration and spatial footprint of the largest labelled extreme events and the effective detection of reported extreme events.*

• l. 235-240: Can you better report about this in the methods section?

*Yes, a paragraph has been added in the method section.*

---

## Author Comment (AC2)

*We thank the reviewer for their comments and suggestions. Please find hereafter our replies in italics shaded in blue. For the sake of clarity, a clear distinction is made between compound dry and hot (CDH) events and the dry and hot extreme event database (Dheed).*

**RC1**: 'Comment on essd-2024-396', Anonymous Referee #1, 16 Nov 2024

**Citation**: https://doi.org/10.5194/essd-2024-396-RC1

The manuscript essd-2024-396 presents a dataset identifying historical compound dry and hot extreme events derived from ERA5 global meteorological data. The study's focus on compound events, which are more damaging than univariate extremes, is timely and valuable for the community. However, several issues require attention to enhance the dataset's reliability and utility.

**Major Concerns:**

1. **Global Trend of Heat Extremes**:

   The data analysis indicates a non-significant trend in heat extreme day numbers globally from 1970 to 2022. This finding contradicts numerous studies showing that heatwaves have become more frequent and severe over time. This discrepancy should be thoroughly investigated, as it undermines the reliability of the dataset.

   *We thank the reviewer for bringing our attention to this result, which also puzzled us. After a careful revision of our workflow, we identified an error in our code, which introduced a monthly bias in the temperature and precipitation data. All the analyses have been run again and the new results show a significant trend in the annual number of hot days.*

2. **Data & Methodology**:

   a. The dataset relies exclusively on ERA5 data. The authors should include a literature review demonstrating that ERA5 is widely accepted for historical drought and heat stress analyses.

   *ERA5 has been widely used in scientific publications related to drought and/or heat. A Google Scholar search on citations of Hersbach et al. 2020 reveals that out of the 19,587 citations, 5,510 contain the keyword drought and 10,500 contain the keywords heat or heatwave. Searching for drought AND (heat OR heatwave) reveals 3,600 citations. https://scholar.google.com/scholar?hl=en&as_sdt=2005&sciodt=0%2C5&cites=18403910731188548420&scipsc=1&q=drought+AND+%28heat+OR+heatwave%29&btnG=*

   *Nevertheless, various studies report limitations regarding precipitations, especially in the tropics.*

   b. The use of the 1% threshold for defining "extreme" events requires justification through references to relevant literature.

*Most studies on extreme events use a threshold of 5% or 10% (resp. 95% or 90%) on monthly anomalies based on a 30 year reference period, corresponding to a return period for a specific location and month of about 20 or 10 years. In this study, we use a longer reference period (74 years) and daily data, which, for a grid cell, leads to an annual average of 3 to 4 days flagged as extremes for a single indicator.*

c. The parameterization G=0.5Rn for ground heat flux is uncommon as this ratio is typically associated with vegetation cover. Now the impact from the surface cover is missing. This choice should be explained or supported with references.

*For the reference evapotranspiration, we adopted the parametrization G=0.5Rn for the nighttime soil heat flux density following Allen et al. 1998 (Equations 45 and 46) and Singer et al., 2021. This value is not the actual nor the potential evapotranspiration but a reference for a well watered grass cover.*

3. **Seasonal Detrending**:

The proposed methodology lacks seasonal detrending, a standard preprocessing step in drought and heat analyses. Without removing seasonal cycles, anomalies are compared to absolute values rather than seasonal baselines. For example, warmer days in spring might qualify as heat stress even if their absolute temperatures are lower than those in summer. Similarly, seasonal cycles in PEI could influence water stress results. This methodological issue weakens the robustness of the findings and should be addressed.

*The rationale behind the choice of not removing the mean seasonal cycle (MSC) is that the primary aim of our database is to analyse the impacts on the vegetation, which are more sensitive to absolute thresholds than to relative ones (see, e.g., Marchin et al. 2022). Especially, in a changing climate with a shifting reference, the MSC should be adapted dynamically. A shift in the start of the growing season or the rainy season would be considered as extreme in terms of anomalies, while the physiological consequences wouldn't necessarily be such.*

4. **Omission and Commission Errors**:

The authors conducted a qualitative literature survey of extreme events captured or missed by the dataset. This evaluation is commendable but could be improved by quantitatively summarizing omission and commission errors. These statistics should also be briefly mentioned in the abstract for clarity.

*We thank the reviewer for this valuable suggestion. However, we think that commission and omission errors computed on 40 reported events and 20 events from the Dheed database might underestimate the actual error. Indeed, on the one hand, the chosen events are spatially large and/or last many days, which is not the case for most of the events in the database. On the other hand, searching for references to small events might be unfeasible without dedicated text mining and web crawling tools, which are outside of the scope of this study.*

*Nevertheless, we will add a sentence in the revised abstract and manuscript stating that: Out of 40 events selected a priori, 38 could be associated with Dheed events. All 10 largest and 10 longest events from Dheed could be linked to events reported in the scientific or grey literature.*

5. **Global Trend Mapping**:

Figure 6 presents trend analyses across continents. A pixelwise global trend map would provide a more detailed spatial representation and is strongly recommended.

*We thank the reviewer for this suggestion. We will add a pixelwise decadal trend map to the revised manuscript. Indeed at the grid cell scale, an annual trend map doesn't make sense because, thankfully, there aren't extremely dry and hot days every year. However, when aggregating the number of extremely dry and hot days over ten years, a significant trend can be observed in many places.*

6. **Intercomparison with Other Indices**:

The dataset should be compared with drought and heat indices from other sources over a long time span to validate its reliability.

*We thank the reviewer for this suggestion. The qualitative validation of the CDH events shows that our indices are suited to detect large events. A more systematic quantitative comparison with indices from other sources could indeed reinforce the confidence in the Dheed database. Under the constraint of the revision deadline, we haven't been able to do this in the time imparted. We would compare the daily PEI with daily SPEI at ICOS sites (EOBS based, Pohls et al 2023 https://doi.org/10.5281/zenodo.7561854; ERA5 based Liu et al 2024 https://doi.org/10.5281/zenodo.8060268).*

**Additional Comments:**

- **Data Citation**: ESSD requires the data URL and citation to be explicitly mentioned in the abstract.

  *The data URL and citation will be added to the revised abstract.*

- **Figure 1**: The terms "ET0" and "ETref" should be unified for consistency.

  *We will use the notation ET0 throughout the manuscript, consistent with Allen et al 1998*

- **Hourly ETref Calculation**: Clarify why θ (daily mean temperature) is used instead of hourly temperature.

  *There was an error in the manuscript. The hourly temperature is used in the calculation. This will be corrected in the revised manuscript.*

- **Missing References**: Two papers by Ima are cited but not included in the reference list.

*The two pieces of software have no author and their references were included in the beginning of the list. They have now been moved to follow the alphabetical order, using the software title as reference entry.*

- **Figure 2**: Clarify the difference between "no extreme" and "10th–90th percentile."

*All values lying between the 10th and 90th percentiles are flagged as no extreme. The tails of the distributions (values smaller than the 10th percentile or greater than the 90th percentile are depicted in white. There was an error in the description of the figure, stating that the white colour shows the centre of the distribution, while the grey colour does. The caption will be modified accordingly in the revised manuscript.*

- **Figure 3**: Add a legend explaining the meaning of the additional colors.

*The following sentence will be added to the revised manuscript:*

*The shades of blue and purple show the accumulation period of the water balance. The darker the shade the longer the accumulation period: a water balance accumulated over 180 days which is below the 1% threshold is rendered in the darkest shade. The 90 day accumulation period is shown in the medium shade. The 30 day accumulation period has the lightest shade.*

- **Table 2**: Specify the units for area and volume.

*The area and volume are not properly area and volume but are proportional to the area and to the area multiplied by the duration of the event in days. The following sentence will be added to the table caption: The area is an adimensional proxy of the spatial area affected by an event obtained by counting the number of voxels in an event multiplied by the cosine of their respective latitude (volume) divided by the number of days between the start and the end of the event (duration). An area of 1 is the size of a 0.1x0.1 degree grid cell at the Equator or about 122 $km^2$.*

- **Data Accessibility**: URLs for the data cubes are not accessible. Please ensure they are active and functional.

*We are sorry that the reviewer was not able to access the data at the provided URL. The zenodo links do work. There was indeed an error in the s3 link, which should have been [https://s3.bgc-jena.mpg.de:9000/deepextremes/v4/ReadMe.md](https://s3.bgc-jena.mpg.de:9000/deepextremes/v4/ReadMe.md). The links to the revised version of the database will be corrected in the revised manuscript.*

---

## Author Response (AR2)

We appreciate the feedback provided by the reviewers and the opportunity to address the suggestions and concerns raised. Please find hereafter our replies in italics shaded in blue.

**Anonymous Referee #1**

nominated 28 Mar 2025, accepted 28 Mar 2025, report 07 Apr 2025 Report #1

The authors have addressed most of my questions. The work is solid, and I appreciate the authors' contribution. However, based on the responses from the previous round, a few concerns remain:

1. Regarding the decision not to remove the seasonal cycle, while it is reasonable that the absolute temperature threshold is more suitable for assessing vegetation health, I still think that relative values of surface wetness—compared to the seasonal average—are more appropriate. If the goal is to capture water stress thresholds, these should vary depending on the growing season, implying a dependency on the seasonal cycle. Vegetation may need different water amount for growing and grown seasons. Without accounting for seasonal variation, most drought events may only be detected during the dry season. Could the authors provide further discussion or clarification?

We acknowledge the importance of accounting for seasonal variations in assessing water stress thresholds. Nevertheless, in the applications foreseen for the Dheed when it was conceived, we are convinced that an absolute local threshold does make sense, highlighting the driest days in absolute independently from the season. We recognise that for regions with a strong seasonal cycle, the detected dry events will always occur in the dry season, which won't correspond to the growing season, especially for grassland ecosystems. For permanent vegetation, however, these extreme dry and hot episodes may have important physiological impacts.

Further justification to this choice has been added to the manuscript at lines 173-175:

In regions with a strong hydrological seasonal cycle, adopting a static threshold will lead to the detection of extreme dryness in the dry season, which does not correspond to the growing season. While annual plants may not be affected by these dry episodes outside the growing season, they may have important physiological effects on permanent vegetation.

2. Some spatial figures include ocean pixels for compound hot and dry events, which seems inappropriate. For instance, why would ocean pixels exhibit drought stress? This issue appears in Figures 2, 8, and 10.

As explained in our methodology, ocean pixels were considered in the detection of the events to ensure spatial connection across water bodies. However, these water pixels were not considered in the final statistics. To avoid confusion, ocean pixels are masked in Figures 8 and 10.

3. The manuscript contains inconsistent font types for certain terms, such as EventCube and YAXArrays.jl. Please ensure consistent formatting throughout.

We thoroughly reviewed the manuscript to ensure consistent formatting, particularly for terms like EventCube and YAXArrays.jl, and made necessary corrections.

**Anonymous Referee #3**

nominated 23 Apr 2025, accepted 23 Apr 2025, report 06 May 2025 Report #2

The authors have developed a dataset for compound dry and hot extreme events based on ERA5 dataset. However, there are also a number of datasets available for CDHEE, like the paper published by Yin et al., 2025, who developed a global compound events detection and visualisation toolbox and dataset.

While we acknowledge the existence of similar publications, the publication cited by Referee #3 came out long after we submitted our initial manuscript and only a few weeks before we submitted our revised manuscript. More importantly, it is a toolbox rather than a database and the dataset provides annual statistics of threshold exceedances based on grid cell level. As a database, our work provides unique contributions by integrating specific methodological innovations and providing a ready-to-use global database of CDHE, including a comprehensive list of detected events. Besides, the terabytes of necessary storage and computing power to create such a large dataset are not available to everyone, and as such it offers easy access to extreme event footprints as basis for an array of scientific studies. The revised manuscript includes a reference to the aforementioned publication and better highlights these distinctions and the added value of Dheed as follows:

[Line 71] Yin et al. (2025) recently published a Compound Events Toolbox and Dataset, which provides annual statistics of threshold exceedance for dry and hot days based on total daily precipitation and maximum daily temperature, but lacks an explicit spatio-temporal delineation of the detected extreme events allowing to browse through individual events

Also, I do not see any sound methodology or rigor in this manuscript.

We believe Referee #3 misunderstood core aspects of the paper, which we elaborate below and clarify in the revised manuscript.

1) The authors highlighted the dataset in daily conditions. But the datasets only provide the CDHEE for 30, 90, 180 days. There are already studies addressed CDHEE in a more shorter period (e.g., Tripathy, et al., 2023)

Referee #3 appears to misunderstand the core concept of the paper. Our dataset does indeed provide daily basis CDHEE, while detecting drought using a non standardized PEI with these three accumulation periods (30, 90, 180 days). The resulting PEI indices are still at daily timescale, as is our event detection as is clearly explained around lines 149-150. We clarified this also in the abstract.

2) This study used an empirical distribution function rather than parametric distributions and claimed the latter method is difficult for many grid cells. This should be further justified. I think parametric distributions would work for SPEI, especially for a long period larger than 30 days.

We appreciate the feedback on our methodological approach. We chose the empirical distribution function due to its flexibility across diverse grid cells. In the revised manuscript, we provide a more detailed justification for this choice and discuss its advantages over parametric distributions, particularly for long-term analyses in Appendix A.

[Line 181-183]

However, in this study, we do not remove the effect of seasonality, which leads to distributions poorly described by a log-logistic function, especially where hydrological seasons are strongly distinct (see Appendix A).

3) Why the lowest 1% of the empirical cumulative distribution is considered as the threshold for an extreme? For SPEI, there are already thresholds to categorise different dry conditions.

As explained in the manuscript, the drought indicators are not standardised and the 1% threshold was selected based on a compromise to capture extreme events effectively while limiting their spatiotemporal extent. See lines 235-240

- 4) if 1% is the threshold for SPEI, how about the threshold for hot?
- 5) The detection for the heatwave is unclear. The thresholds for heatwave, even for 1% probability, is identified for all hot season together, or they are detected day by day (i.e., each day has a threshold) like Man and Yuan (2021).

Our methodology for heatwave detection indeed also involves a 1% threshold, computed locally but for all days of the year. Therefore the winter heatwaves are not detected. This is explained and justified at lines 160-180 and now also appears in the abstract.

**Reference:**

Yin, C., Ting, M., Kornhuber, K. et al. CETD, a global compound events detection and visualisation toolbox and dataset. Sci Data 12, 356 (2025). https://doi.org/10.1038/s41597-025-04530-x

Ma, F., & Yuan, X. (2021). More persistent summer compound hot extremes caused by global urbanization. Geophysical Research Letters, 48, e2021GL093721. https://doi.org/10.1029/2021GL093721

K.P. Tripathy, S. Mukherjee, A.K. Mishra, M.E. Mann, & A.P. Williams, Climate change will accelerate the high-end risk of compound drought and heatwave events, Proc. Natl. Acad. Sci. U.S.A. 120 (28) e2219825120, https://doi.org/10.1073/pnas.2219825120 (2023).

**Additional edits to the manuscript:**

Press articles, Reports and software packages without a DOI have been removed from the references' list and their webpages entered as footnotes.

---

## Author Response (AR3)

We thank the reviewers for their feedback and wish to address their remaining concerns. Please find hereafter our replies in italics shaded in blue.

Anonymous Referee #3 nominated 12 Aug 2025, accepted 13 Aug 2025, report 26 Aug 2025 Report #2

I am still concerned about the definition of CHD. The CHD is defined as Tmax exceeding its threshold, while PEI30, PEI60, or PEI180 falls below the 1% threshold. However, this raises two questions:

We acknowledge the reviewer's concern regarding the overestimation and underestimation of compound hot and dry events. However, we are convinced that our approach is a good compromise, even if not perfect. Our method successfully detected and matched the referenced historical events, demonstrating strong agreement and confirming that the approach is both realistic and applicable.

1) A daily Tmax exceeding its 99% threshold indicates only a one-day hot weather event. In contrast, PEI30, PEI60, or PEI180 represents the averaged water balance over the past 30, 60, or 180 days, which reflects a long-term phenomenon. This creates a mismatch when combining hot and dry conditions. For example, consider an extreme case: one day records an extremely high temperature accompanied by a heavy storm. However, if the previous 29 days had absolutely no rainfall, the event may still be detected as CHD due to the preceding dry conditions. This may overestimate the frequency of CHD events, as it may capture situations where extreme heat and dryness do not truly overlap in time.

Drought and heatwaves typically occur at different time scales. The drought conditions are always assessed on longer time scales, typically months or years. Flash drought of less than a month have however also been studied, but they are not the focus of this study as the major effects on vegetation productivity occur through longer-term water deficits. A heavy storm occurring on a single day will not necessarily drastically change the underlying drought conditions for the vegetation, especially not for (deeper rooting) trees and not necessarily proportional to the intensity of precipitation, as heavy precipitation will most probably incur a lot of runoff which won't be available for uptake to the vegetation (However, this out of this study's scope). Drought is not necessarily defined by the complete absence of precipitation but by the relative deficit thereof. Also, a heavy storm will most probably make the maximum temperature drop on the day of the precipitation event or the next (see for example Fig.3). The heat criterion might have been a combination of both Tmax and Tmin to avoid those cases completely. We did give this issue thought in our experimental design, which is why our criteria for the definition of the CDH event require that at least three consecutive days are hot and dry, lowering the chance of encountering such cases.

2) If the hot event does not occur exactly on the 30th day, but instead happens within the preceding 1st to 29th day, how should such an event be characterized? Such an event might not be captured under the current definition

Indeed, our detection method requires that the average reference evaporative stress (PEI) over the target day and the previous 29 days (respectively 89 or 179) be under the 1% of its distribution. Such an accumulation period is common in drought definition: a few relatively dry days could make a flash drought, but this is not the focus of this study. Again, Figure 3 illustrates this well: a heatwave occurred at the end of June 2021 near Lytton, British Columbia, while none of the PEI was under its threshold. PEI\_30 only became extreme around the 15th of July. A few hot days occurred at the beginning of August, but are not part of a labelled event, because the heatwave didn't last at least three consecutive days. Only from the 11th of August were the three criteria met: extreme heat, extremely low PEI (both 30 and 90 in this case) and at least three consecutive days of both previous criteria. Those four days are hence part of the labelled event 130727 in our database, whose statistics can be retrieved with the following Julia code snippet.

```julia

using YAXArrays, Zarr, Dates, DataFrames

import CSV

labelpath =

"https://s3.bgc-jena.mpg.de:9000/deepextremes/v4/mergedlabels\_ranked\_pot0.01\_ne0.1\_cmp\_S1\_ T3 1950 2023.zarr"

labels = Cube(zopen(labelpath))

mylabel = labels[latitude=Near(50.23), longitude=Near(-121.59+360), Ti=At(

Date("2021-08-11"))].data[:]

tmppath =

download("https://s3.bgc-jena.mpg.de:9000/deepextremes/v4/MergedEventStats\_landonly\_int.csv") cdhstats = CSV.read(tmppath, DataFrame)

show(stdout, MIME("text/csv"), filter(:label => x-> x==mylabel, cdhstats))

• • • •

"label", "start\_time", "end\_time", "longitude\_min", "longitude\_max", "latitude\_min", "latitude\_max", "t 2mmax\_mean", "t2mmax\_min", "t2mmax\_max", "pei\_30\_mean", "pei\_30\_min", "pei\_30\_max", "pei\_9 0\_mean", "pei\_90\_min", "pei\_90\_max", "pei\_180\_mean", "pei\_180\_min", "pei\_180\_max", "heat", "drought30", "drought90", "drought180", "compound", "land\_share", "inth", "intd30", "intd90", "intd180", "volume", "duration", "area"

130727,"2021-08-11T00:00:00","2021-08-15T00:00:00",237.75,244.5,49.25,52.5,28.5420693038134 9,20.479685149327395,33.92608105757847,-1.7460892921990534,-2.6781122637298655,-0.28576 391032335413,-0.9112429926574818,-2.37201260008702,0.9771429682567515,-0.0621664542635 9538,-1.493845561964741,1.8248299548958824,100.0,59.82,70.94,49.85,100.0,100.0,165.8145904 3406002,73.9858421244105,119.14148039770703,96.5862652613458,320.76162123680115,"5 days",64.15232424736023

| label         | 130727              |
|---------------|---------------------|
| start_time    | 2021-08-11 00:00:00 |
| end_time      | 2021-08-15 00:00:00 |
| longitude_min | 237.75              |
| longitude_max | 244.5               |

| latitude_min | 49.25          |
|--------------|----------------|
| latitude_max | 52.5           |
| t2mmax_mean  | 28.5420693     |
| t2mmax_min   | 20.47968515    |
| t2mmax_max   | 33.92608106    |
| pei_30_mean  | -1.746089292   |
| pei_30_min   | -2.678112264   |
| pei_30_max   | -0.2857639103  |
| pei_90_mean  | -0.9112429927  |
| pei_90_min   | -2.3720126     |
| pei_90_max   | 0.9771429683   |
| pei_180_mean | -0.06216645426 |
| pei_180_min  | -1.493845562   |
| pei_180_max  | 1.824829955    |
| heat         | 100            |
| drought30    | 59.82          |
| drought90    | 70.94          |
| drought180   | 49.85          |
| compound     | 100            |
| land_share   | 100            |
| inth         | 165.8145904    |
| intd30       | 73.98584212    |
| intd90       | 119.1414804    |
| intd180      | 96.58626526    |
| volume       | 320.7616212    |
| duration     | 5 days         |
| area         | 64.15232425    |

**As a reminder, this is how the caption of Figure 3 reads:**

Heat and drought indicators during a reported compound dry and hot extreme event in the summer of 2021 in British Colombia. Panels show (a) the maximum daily temperature, (b) the daily precipitation and reference evapotranspiration, (c) the three drought indicators (PEI) and (d) the Discrete Extreme Occurrences (DEO). A first heatwave starting 25-06-2021 is not associated with a drought. A second (30-07-2021) and third (03-08-2021) heatwaves are associated with extremely dry conditions but last only two days each. A fourth heatwave starting 11-08-2021 and lasting four days is associated with extremely dry conditions (PEI\_30 and PEI\_90) and is hence part of a labelled event from the proposed database.

We've added the following to the existing discussion (current manuscript excerpts in squared brackets):

[The global event detection of compound dry and hot extreme events faces the difficulty of dealing with processes that happen at different time scales. ]

[...]

[The framework presented here concentrates on detecting and labelling droughts and heatwaves and their compound occurrence based on daily meteorological data.]

Our approach relies on daily data and defines CDH at daily scale, but using 30, 90 and 180 days accumulation periods for assessing dry conditions. Hence, a DEO combines heat on the day with accumulated water stress, allowing to reconcile the differing time scales of drought and heat. Flash droughts are not a focus of this study. Only three consecutive DEOs make it to a labelled CDH, alleviating the overestimation of CDH events.

[The resulting labelled CDH events can be used to analyze trends at regional, continental and global scales and to drive further research into the impacts of such events on ecosystems, specific species or society.]

---

## Author Response (AR4)

We thank the reviewers for their feedback and wish to address their last comments. Please find hereafter our replies in italics shaded in blue.

Anonymous Referee #3 nominated 20 Sep 2025, accepted 20 Sep 2025, report 22 Oct 2025 Report #2

My concerns have been clarified and resolved. I have no more questions

We thank the reviewer for appreciating our efforts to clarify our work.

Referee #4: De Luca, Paolo paolo.deluca@bsc.es nominated 04 Oct 2025, accepted 04 Oct 2025, report 07 Oct 2025 Report #1

In the Introduction and/or Discussion you may consider adding:

- 1) https://agupubs.onlinelibrary.wiley.com/doi/full/10.1029/2022GL102493
- 2) <a href="https://esd.copernicus.org/articles/11/793/2020/esd-11-793-2020.html">https://esd.copernicus.org/articles/11/793/2020/esd-11-793-2020.html</a>

Where 1) uses a different approach for computing dry extremes, i.e. not based on percentile thresholds but on SPEI/SPI thresholds <= -1

and 2) uses a compound dynamical system approach based on daily temperature and precipitation.

We thank the reviewer for suggesting to include references to these two relevant publications. We think that the relevance of (1) to our work is not necessarily in the value of the threshold used but rather on the results of the analysis and on the method used to reconcile the different time scales of heat and drought. We have added the following statements in the introduction and the discussion (highlighted in red):

**[Introduction]**

A typology to guide studies on those types of occurrences has recently been proposed (Zscheischler et al., 2020). Analysing model results and future emissions scenarios from 1950 to 2100, De Luca and Donat (2023) showed that "hot, dry, and compound hot-dry extremes are projected to increase over large parts of the globe by the end of the 21st century" and that "dry extreme changes are sensitive to the index used". Compound climate extremes often have more detrimental effects on vegetation growth than univariate extremes (Yang et al., 2023; Bastos et al., 2023).

[...]

Various indicators have been developed to characterize drought conditions. The commonly used Standard Precipitation Evaporation Index (SPEI) is a "multi-scalar drought index used to determine the onset, duration and magnitude of drought conditions" (Vicente-Serrano et al., 2010). It is generally calculated from monthly climate data, which then require adjustments to reconcile the monthly time scale of the drought indicator with the daily time scale of the heat indicator. De Luca

and Donat (2023) converted SPEI monthly time series into daily time series by setting the daily values to the same value over all days in a month. Some authors have used the SPEI with daily data to characterize drought dynamics at a finer temporal resolution (Wang et al., 2021).

[...]

**[Discussion]**

[...]

The compound nature of multi-hazard extreme events could be better apprehended with multivariate distributions. For example, standard multivariate normal kernel has been shown to outperform univariate extreme event detection on synthetic data (Flach et al., 2017) and successfully applied on real Earth system data to detect anomalies (Flach et al., 2021). De Luca et al. (2020) proposed a method based on dynamic systems theory for characterizing dry-hot and wet-cold compound events in terms of the coupling between precipitation and temperature fields, allowing to relate long-term changes in compound events to their underlying physical drivers.

[...]